# CD4$^+$ helper T cells endow cDC1 with cancer-impeding functions in the human tumor micro-environment

Xin Lei[1,2], Indu Khatri [1,7], Tom de Wit[1,2,7], Iris de Rink[3], Marja Nieuwland[3], Ron Kerkhoven[3], Hans van Eenennaam[4], Chong Sun[5], Abhishek D. Garg [6], Jannie Borst [1,2,8] ✉ & Yanling Xiao [1,2,8] ✉

Despite their low abundance in the tumor microenvironment (TME), classical type 1 dendritic cells (cDC1) play a pivotal role in anti-cancer immunity, and their abundance positively correlates with patient survival. However, their interaction with CD4$^+$ T-cells to potentially enable the cytotoxic T lymphocyte (CTL) response has not been elucidated. Here we show that contact with activated CD4$^+$ T-cells enables human ex vivo cDC1, but no other DC types, to induce a CTL response to cell-associated tumor antigens. Single cell transcriptomics reveals that CD4$^+$ T-cell help uniquely optimizes cDC1 in many functions that support antigen cross-presentation and T-cell priming, while these changes don't apply to other DC types. We robustly identify "helped" cDC1 in the TME of a multitude of human cancer types by the overlap in their transcriptomic signature with that of recently defined, tumor-infiltrating DC states that prove to be positively prognostic. As predicted from the functional effects of CD4$^+$ T-cell help, the transcriptomic signature of "helped" cDC1 correlates with tumor infiltration by CTLs and Thelper(h)−1 cells, overall survival and response to PD-1-targeting immunotherapy. These findings reveal a critical role for CD4$^+$ T-cell help in enabling cDC1 function in the TME and may establish the helped cDC1 transcriptomic signature as diagnostic marker in cancer.

Tumor-antigen loaded dendritic cells (DC) have been used as vaccines in hundreds of cancer immunotherapy trials, but not given significant treatment benefits[1]. In most of these trials, monocyte-derived (mo)DC were used, because these can easily be generated in vitro[2]. Whereas moDC differentiate from monocytes, particularly under inflammatory conditions, three acknowledged DC lineages develop at steady-state from progenitors: plasmacytoid (p)DC, classical (c)DC1 and cDC2[3–5]. Recent single cell studies have identified further heterogeneity within these DC subsets, but this likely reflects different transitory cell states,

rather than interspecies conserved lineages as governed by dedicated transcription factors[5]. It has been proposed that the unsatisfactory performance of moDC in cancer immunotherapy is due to their suboptimal intrinsic capacity to induce T-cell responses[5,6]. From extensive studies in mouse, it is clear that the cDC1 and cDC2 lineages, that are discerned into migratory and lymph node-resident subpopulations play a key role in initiating T-cell responses[4,5].

CD4$^+$ and CD8$^+$ T-cell priming in secondary lymphoid organs is the resultant of successive, chemokine-guided interactions with different

[1]Department of Immunology, Leiden University Medical Center, Leiden, The Netherlands. [2]Oncode Institute, Leiden University Medical Center, Leiden, The Netherlands. [3]Genomics Facility, The Netherlands Cancer Institute, Amsterdam, The Netherlands. [4]Aduro Biotech Europe B.V, Oss, The Netherlands. [5]Immune Regulation in Cancer, German Cancer Research Center, Heidelberg, Germany. [6]Laboratory of Cell Stress & Immunity, Department of Cellular & Molecular Medicine, KU Leuven Leuven, Belgium. [7]These authors contributed equally: Indu Khatri, Tom de Wit. [8]These authors jointly supervised this work: Jannie Borst, Yanling Xiao. ✉e-mail: j.g.borst@lumc.nl; y.xiao@lumc.nl

DC types[7]. First, CD4[+] and CD8[+] T-cells are activated independently by migratory cDC2 and cDC1, respectively that bring antigen from peripheral tissue to draining lymph nodes. When sufficient pro-inflammatory signals reach the lymph node, these activated CD4[+] and CD8[+] T-cells undergo a second step of priming[8]. Herein, they interact with the same cDC1 that presents their antigens of interest. The CD4[+] T-cell then "licenses" the cDC1, largely via CD40 signaling, to give the CD8[+] T-cell specific instructions for proliferation and CTL effector- and memory differentiation[7].

Among human and mouse DC types, the cDC1 excels in antigen cross-presentation[9,10]. In this process, endocytosed proteins or cell debris is digested into peptides that are presented in Major Histocompatibility Complex (MHC) class I molecules to CD8[+] T-cells[11]. In this way, antigens derived from dead tumor cells or virus infected cells can evoke a cytotoxic T lymphocyte (CTL) response. Accordingly, the cDC1 plays a key role in the CTL response against cancer[12]. The cDC1 promotes the CTL response both in lymph nodes and the TME[13] and at both locations, the cDC1/CTL interface is the target of PD(L)−1 immune-checkpoint blockade (ICB)[14]. Mouse studies additionally implicate cDC1 in relaying CD4[+] T-cell help to CD8[+] T-cells, which optimizes CTL effector functions against cancer[7,15,16]. Whether the cDC1 is critical for delivering "help" signals to CTL in human is unknown.

Here we show, in a human in vitro setting, that only cDC1, but not pDC, cDC2 or moDC can relay CD4[+] T-cell help for CTL priming to cell-associated tumor antigens. By single cell mRNA sequencing (scRNA-seq) and flow cytometry, we identify among these DC types the cDC1 as the major responder to CD4[+] T-cell help signals and define the nature of cDC1-licensing at the molecular level. The licensed/helped cDC1 uniquely acquires a gene expression/protein signature highlighting antigen (cross)presentation and specific costimulation-, cytokine- and chemokine features that indicate an optimized capacity to induce T-cell responses. These features are more explicitly and sometimes uniquely instilled in cDC1 by CD4[+] T-cell help as compared to pattern recognition receptor (PRR) stimulation. We discover the similarity of the "help" transcriptomic signature with recently identified DC3[17] and DC_S3 states[18] in the TME, of which the latter is positively prognostic in 16 different tumor types, derived from thousands of cancer patients.

This study reveals how CD4[+] T-cell help optimizes human cDC1 function for inducing a CTL response against cancer cells, which provides strong arguments to engage CD4[+] T-cell help in cancer immunotherapy. We find the signature of "helped/licensed" cDC1 in the TME of a large range of human cancers, which argues that CD4[+] T-cell help for the CTL response can take place within T cell-infiltrated tumors. The correlation of the cDC1 "help" signature with CTL- and Th1 cell infiltration in the TME, good prognosis and response to checkpoint immunotherapy argues for its utility in clinical diagnostics.

## Results

### scRNA-seq reveals unique ability of cDC1 to respond to CD4[+] T-cell help

pDC, cDC1, cDC2 and moDC were purified from peripheral blood of healthy donors by flow cytometric sorting based on cell surface markers[19] (Supplementary Fig. 1a, b) and validated by transcriptome analysis. Hierarchical clustering indicated the relationships between the DC subsets, with pDC separating from the other subsets and ex vivo moDC being more closely related to cDC2 than to cDC1. The transcriptome highlighted the common element of MHC class II pathway expression and subset-specific TLR expression (Supplementary Fig. 1c, d). Subset discrimination was further validated by 114 transcripts encoding CD markers as defined by Human Protein Atlas, indicating e.g. CD14 expression exclusively in ex vivo moDC and *XCR1* and *CLEC9A* expression exclusively in cDC1 (Supplementary Fig. 1e, f). Comparison with single cell transcriptome data from Villani et al.[20] confirmed the distinction between the pDC, cDC1 and cDC2 subsets and demonstrated

a monocyte-related signature in ex vivo moDC (Supplementary Fig. 2a). The latter was confirmed by comparison with the CD14[+]CD163[+] DC signature derived from another single cell dataset[21] (Supplementary Fig. 2b). Furthermore, the ex vivo moDC had high expression of the signature transcripts published for in vitro generated moDC[22] (Supplementary Fig. 2c). In the manuscript, the term moDC will refer to the ex vivo moDC as here described, unless otherwise specified. The DC subsets used in current study partially discerned themselves by pathways related to phagocytosis, receptor signaling and antigen presentation, especially cDC1 (Supplementary Fig. 2d).

To understand the cellular and molecular mechanism of DC licensing by CD4[+] T-cells, we determined the response of all four DC subsets to activated CD4[+] T-cells by scRNA-seq. In separate samples, the purified DC subsets were co-cultured in equal numbers with naïve or anti-CD3/CD28-activated CD4[+] T-cells overnight. By activating the CD4[+] T cells via the TCR/CD3 complex, we mimic recognition of the MHC/peptide complex. CD28 stimulation further enhances the signaling events that amongst others result in cytoskeletal rearrangement and synapse formation that allows receptor-ligand communication between the activated T cell and the DC[23]. Hashtag scRNA-seq[24] was designed to evaluate the transcriptomic profile of the DC in each of the 8 different samples in one analysis (Supplementary Fig. 3a). The cells in the DC-CD4[+] T-cell cultures were all labeled with antibody to ubiquitous β2m, that was conjugated to 8 different hashtag oligonucleotides (HTO) to identify each sample, and with an HTO-conjugated antibody to CD3 to identify T cells. By sequencing the HTO alongside the cellular transcriptome, each cell could be assigned to its original sample. For classifying each barcode as a "positive" singlet HTO, "multiplets" and "negatives" were excluded. CD4[+] T-cells were excluded for DC fraction analysis based on *CD3D* mRNA expression and the HTO against CD3. Singlet HTO 1–8 populations were clearly identified (Supplementary Fig. 3b, c). Transcriptome-based clustering of the classified singlets enabled detection of four DC subsets, confirmed by their HTO identity (Supplementary Fig. 3d), although the number of pDC recovered and passing quality control was very small. Gene expression profiles of pDC and moDC co-cultured with either activated- or naïve CD4[+] T-cells were not different. However, 577 differentially expressed genes (DEGs) were detected in cDC1 co-cultured with activated- versus naive CD4[+] T-cells and 87 DEGs were detected in cDC2 under the same comparative conditions (Fig. 1a, Supplementary Fig. 4a, b; Supplementary Data 1).

### "Helped" cDC1 increase expression of molecular pathways that support CTL cross-priming

To understand the molecular mechanisms that activated CD4[+] T-cells induce in human cDC1, the 577 DEG in the cDC1 were subjected to gene ontology (GO) and gene set enrichment analysis (GSEA). We uncovered that pathways important for T-cell priming were activated in "helped" cDC1, such as those connected to DC viability, antigen processing and (cross-)presentation, chemokine-guided DC- and T-cell recruitment/migration, DC maturation, interleukin signaling and T-cell differentiation (Fig. 1b, Supplementary Fig. 4c). At the mRNA level, activated CD4[+] T-cells induced in cDC1 a significant upregulation of costimulatory molecules CD40, CD83 and CD86, but also PD-L1 (CD274), specific cytokines (IL-15, IL-32), the chemokine receptor CCR7 and the chemokines CXCL9/10/11, diverse components of the MHC class I antigen presentation pathway including HLA-A/B/C, core proteasome subunits (PSMB8/9, PSMA2) and the transporter associated with antigen processing (TAP1/TAP2) (Fig. 1c). The top 100 upregulated DEGs of the cDC1 "help" gene expression signature are shown in Fig. 1d.

Flow cytometry analysis confirmed increased expression for many of these molecules at the protein level after co-culture with activated but not naïve CD4[+] T-cells specifically in cDC1 but not in pDC, cDC2 or

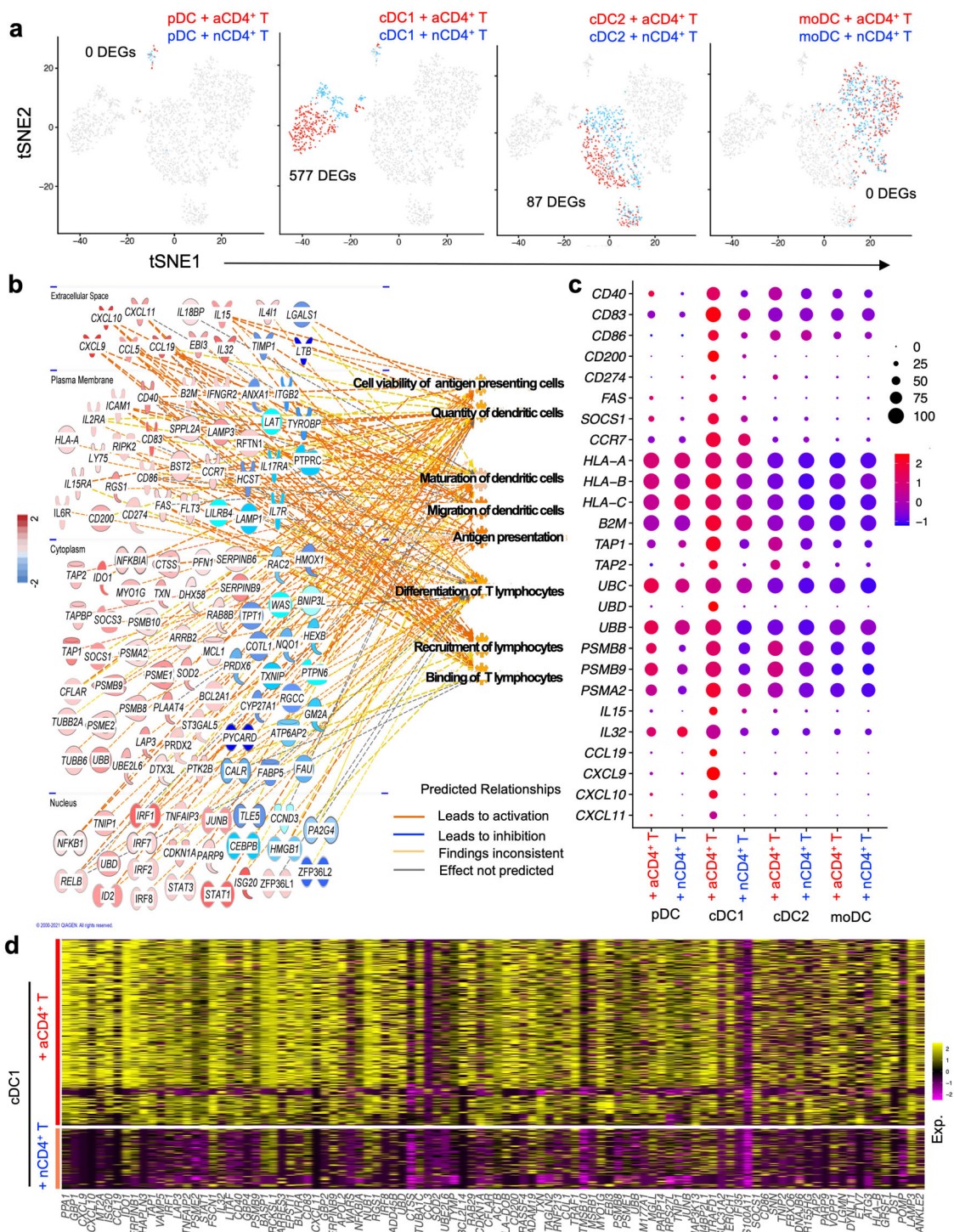

**Fig. 1 | "Helped" cDC1 increase expression of molecular pathways that support CTL cross-priming.** CD11c⁻CD303⁺ pDC, CD11c⁺CD141⁺ cDC1, CD11c⁺CD1c⁺ cDC2 and CD11c⁺CD14⁺CD206⁺ moDC were flow cytometrically sorted from human PBMC as outlined in Supplementary Fig. 1a. Cells were co-cultured with activated- or naive CD4⁺ T-cells overnight, then stained with antibody to β2m- conjugated to hashtag oligonucleotides (HTO) 1-8 and oligo-tagged antibody to CD3. After extensive washing steps, HTO 1-8 tagged samples were pooled in equal proportion and loaded on a 10X Genomics platform. **a** tSNE plots highlighting the mRNA expression profiles of pDC, cDC1, cDC2 and moDC subsets individually. Red color indicates DC co-cultured with activated (a)CD4⁺ T-cells. Blue color indicates DC co-cultured with naive (n)CD4⁺ T-cells. **b** GO biological process analysis using Ingenuity Pathway Analysis (IPA) using the 577 DEGs of the cDC1 "help" signature. **c** Dot plot depicting transcript levels and percentage of cells expressing genes related to key pathways of "antigen processing-cross presentation", "DC maturation/migration" and "T cell differentiation/recruitment" as identified in the cDC1 "help" signature. **d** Heatmap revealing top 100 upregulated DEGs in the cDC1 "help" signature as derived from comparing activated CD4⁺ T-cell treated cDC1 (HTO2) versus naïve CD4⁺ T-cell treated cDC1 (HTO6).

moDC and demonstrated additionally the upregulation of costimulatory ligands CD70 and CD80 and MHC class II molecule HLA-DR (Fig. 2a, Supplementary Fig. 5). Although the gene expression changes in cDC1 induced by activated CD4[+] T-cells and pattern recognition receptor (PRR) stimuli partially overlapped, "helped" cDC1 had higher expression of most of the abovementioned molecules at the protein level and uniquely upregulated CCR7, CXCL9/10 and diverse components of the MHC class I and -II antigen presentation pathway (Supplementary Fig. 6a–d). This validation indicated that among human DC subsets, the cDC1 shows a unique response to CD4[+] T-cell help.

## Human cDC1 is superior to other DC types in relaying CD4[+] T-cell help for anti-tumor CTL priming

To test the ability of the DC subsets in relaying CD4[+] T-cell help for CD8[+] T-cell priming, we established an in vitro tumor antigen-specific CTL priming platform using the primary DC subsets, CD4- and CD8[+] T-cells purified from human blood (Fig. 2b). CD8[+] T-cells were retrovirally transduced to express a T-cell receptor (TCR) specific for MART-1$_{26-35}$ peptide in the context of HLA-A2[25] (Supplementary Fig. 7a). MART-1/Melan-A (MLANA) is a melanocyte-specific protein that is well-studied as CTL target[26]. The transduced T-cells had a CD45RA[+]CD45RO[-/low]CD62L[+] CCR7[+]CD27[+]CD28[+]CD95[+]CXCR3[+] phenotype (Supplementary Fig. 7b, c), indicating a stem cell-like memory T cell (T$_{SCM}$) state[27]. T$_{SCM}$ cells have not yet undergone effector differentiation, according to transcriptome- and epigenetic analysis[27], which was important for our in vitro priming assay. Transduction efficiency, as determined by MHC tetramer staining, was about 40% (Supplementary Fig. 7d, e). To test the impact of CD4[+] T-cell help for CTL priming, we added either naïve or activated CD4[+] T-cells to the DC, and CD8[+] T-cells were labeled with the fluorescent dye Cell Trace Violet (CTV) to monitor proliferation. By using MART-1$_{15-40}$ long peptide or dead melanoma cell debris, we created antigen cross-presentation settings, wherein exogenous (cell-associated) protein is processed to generate MART-1$_{26-35}$ peptide for presentation in the context of HLA-A2. The HLA-A2[+] Mel 526 or HLA-A2[−] Mel AAT melanoma cell lines expressing MART-1[26] (Supplementary Fig. 7f) were used as cell-associated antigen source. For this purpose, cells were induced to undergo apoptosis by treatment with death receptor ligands (Supplementary Fig. 7g, h). During antigen loading, DC were additionally activated with a mixture of PRR stimuli (poly I:C, LPS and R848), given that different DC subsets express different PRR (Supplementary Fig. 1d).

These cultures reliably reported MART-1$_{26-35}$-specific T-cell priming based on CTV dilution and Granzyme B production in TCR-transduced CD8[+] T-cells (Supplementary Fig. 8a). In the cross-presentation setting with MART-1$_{15-40}$ long peptide, CTL priming was observed when DC were helped by activated CD4[+] T-cells (Fig. 2c). A response was observed with cDC1, cDC2 and moDC, but the "helped" cDC1 was the most potent one in priming MART-1$_{26-35}$-specific CD8[+] T-cell response, both in terms of proliferation (Fig. 2c) and Granzyme B induction (Supplementary Fig. 8b, c). In the cross-presentation setting with Mel cell debris, only "helped" cDC1 could induce MART-1$_{26-35}$-specific CD8[+] T-cell proliferation (Fig. 2d, Supplementary Fig. 8d, e) and Granzyme B production (Fig. 2e, Supplementary Fig. 8f, g), whereas pDC, cDC2 and moDC failed to do so (Fig. 2d, e, Supplementary Fig. 8e, g). Strikingly, proliferation and Granzyme B production were now also observed in HLA-A2/MART-1$_{26-35}$ tetramer-negative CD8[+] T-cells (Fig. 2d, Supplementary Fig. 8h–i), suggesting priming of CD8[+] T-cells specific for other antigens than MART-1$_{26-35}$. Apart from Granzyme B, other proteins indicating CTL effector differentiation were also upregulated most explicitly in cDC1 as a result of CD4[+] T cell help (Supplementary Fig. 9a–f). In the same experiments, we found that it was irrelevant for the outcome whether the melanoma cell lines that acted as MART-1 antigen donor expressed HLA-A2, in agreement with the notion that the

HLA-A2[+] DC crosspresented the antigen. This was corroborated by the fact that no MART-1$_{26-35}$-specific CD8[+] T-cell proliferation was induced in presence of either dead Mel 526- or dead Mel AAT cells but in absence of DC (Supplementary Fig. 9g, h). In vitro generated moDC (Supplementary Fig. 10a, b) essentially behaved like our ex vivo isolated moDC in that they did not specifically respond to activated CD4[+] T-cells as compared to naïve CD4[+] T-cells (Supplementary Fig. 10c) and did not relay help for tumor-cell associated antigen cross-priming as compared to the helped cDC1 (Supplementary Fig. 10d, e). These data indicate that among the human DC subsets we interrogated, the cDC1 preferentially responds to CD4[+] T-cell help by optimizing its CTL priming ability, particularly in a setting of cross-presentation of (tumor) cell-associated antigen.

In the mouse, CD40-induced expression of costimulatory ligand CD70 on DC and resulting engagement of its receptor CD27 on CD8[+] T-cells was shown to be an important aspect of CD4[+] T-cell help for the CTL response[15,28,29]. As CD70 was uniquely upregulated in human cDC1 upon activated CD4[+] T-cell stimulation (Fig. 2a, Supplementary Fig. 5g), we tested whether CD70 on human cDC1 performed the same function as it does in mouse. In the cross-presentation setting with MART-1$_{15-40}$ long peptide in the (Fig. 3a), the MART-1 specific CD8[+] T-cell response was significantly reduced by antibody-based CD70 blockade, both in terms of proliferation (Fig. 3b–d) and Granzyme B induction (Fig. 3c, e). These results indicate that also in human, CD70/CD27 co-stimulation at the cDC1 interface is important for CTL cross-priming in response to CD4[+] T-cell help. Overall, we conclude that contact with activated CD4[+] T-cells programs cDC1 to boost all molecular programs that are important to induce a CTL cross-priming.

## Previously identified tumor-infiltrating DC states share features specifically with cDC1 and "helped" cDC1

In scRNAseq analyses of different cancer types from both mouse and human, specific tumor-infiltrating DC types or states were identified that are conserved between mouse and human, and on basis of their mRNA expression signatures termed DC3[30], LAMP3[+] DC[31] and mature regulatory (mreg)DC[32]. A recent meta-analysis of these data describes the immune infiltrates of five different human solid cancer types[17]. Besides pDC, cDC1, cDC2 and moDC that represent lineages present in blood, lymphoid tissues and tumor, the DC3 was found exclusively in tumor tissue[17]. The DC3 is considered a cell state rather than a lineage and includes LAMP3[+] DC and mregDC. Interestingly, we found that the 269 upregulated genes in the cDC1 "help" signature (Supplementary Data 1) comprised 85% of DC3 signature genes (Fig. 4a, b) and 61% of mregDC signature (Supplementary Fig. 11a, b). Moreover, expression of DC3 as well as mregDC signature genes was only revealed in cDC1, but not in pDC, cDC2 or moDC (Fig. 4b, Supplementary Fig. 11b). We also cross-compared our scRNAseq data with a mature DC signature[33] revealed in a variety of tumors with an IL-32[hi] TME that is associated with CD8[+] and Th1-type T-cell and cDC1 infiltration[34]. Nearly 50% of the mature DC signature in the IL-32[hi] TME was included in the cDC1 "help" signature (Supplementary Fig. 11c) and this signature was also only enriched in cDC1, but not in pDC, cDC2 and moDC (Supplementary Fig. 11d).

A novel bioformatic method ("EcoTyper") has recently identified in bulk RNAseq data 12 major cell lineages and 69 defined cell states in the TME of 6475 tumors across 16 different cancer types. The findings were validated by analysis of scRNA-seq from 7 databases across 4 cancer types[18]. In this study, tumor-infiltrating DC were specified into 8 states according to their gene expression profiles (Supplementary Data 1). Two DC states, S1 and S3 that are associated with longer overall survival (OS)[18], were revealed in cDC1, but not in pDC, cDC2 or moDC (Fig. 4c, Supplementary Fig. 12a). Other DC states that mostly associated with shorter OS[18], were hardly revealed in cDC1 (Fig. 4c, d). Specifically, the cDC1 "help" signature shared 30% of transcripts with

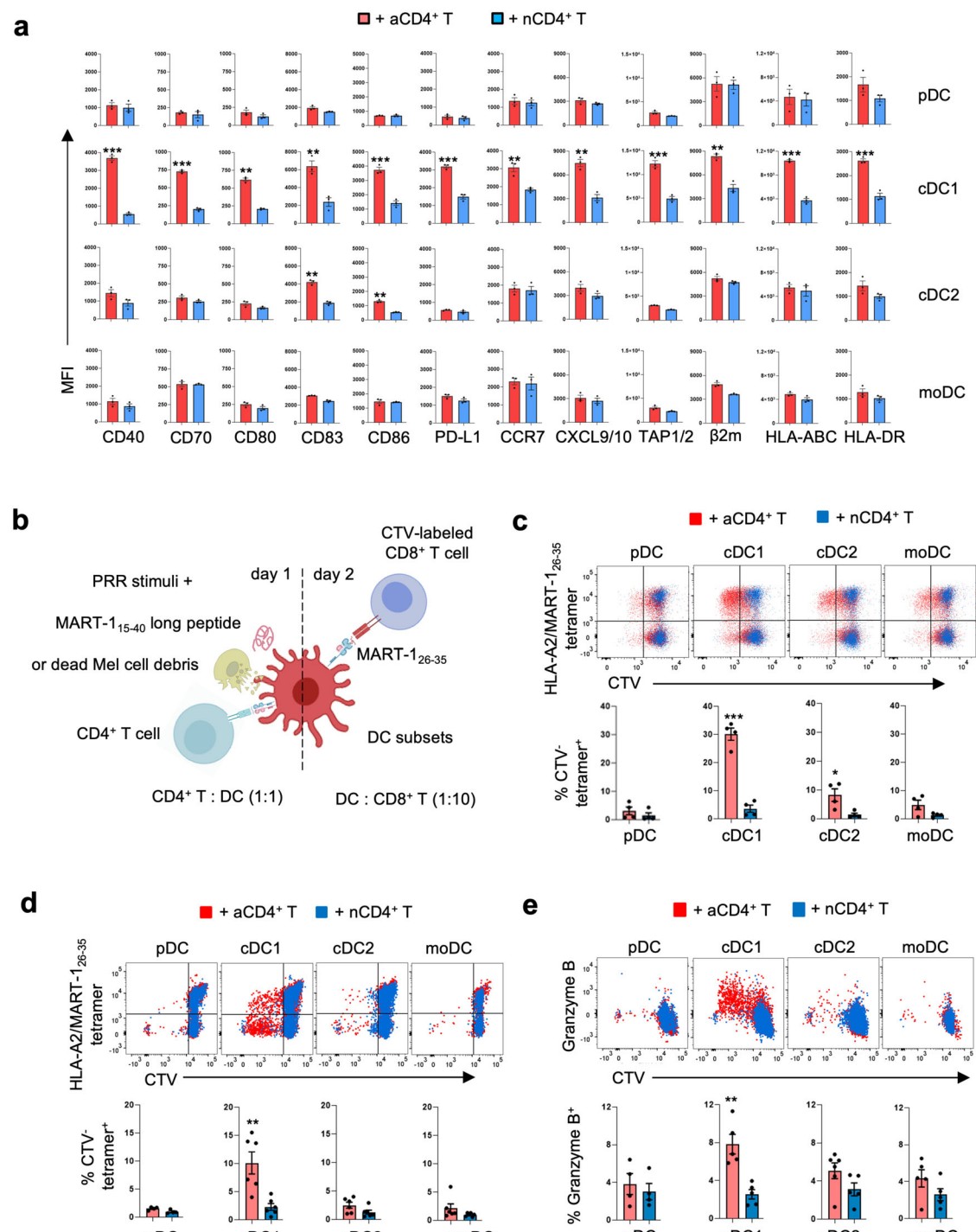

**Fig. 2 | Human cDC1 is superior to other DC types in relaying CD4+ T-cell help for the anti-tumor CTL response. a** The pDC, cDC1, cDC2 and moDC subsets were flow cytometrically isolated and co-cultured with activated (a)- or naive (n)CD4+ T-cells for 12 h as outlined in Supplementary Fig. 5 Next, key molecules of the cDC1 "help" signature were analyzed by flow cytometry. Median Fluorescence Intensity (MFI) quantifications are shown for indicated markers expressed by DC subsets under indicated conditions. Data were pooled from three independent experiments ($n = 3$) with technical duplicates. **b** Schematic depiction of the tumor antigen-specific CTL priming system. On day 1, pDC, cDC1, cDC2 and moDC from HLA-A2+ healthy donors were incubated with activated (a)- or naïve (n)CD4+ T-cells, loaded with MART-1$_{15-40}$ long peptide or dead MART-1+ melanoma cell debris and a mixture of PRR stimuli. On day 2, $T_{SCM}$ phenotype CD8+ T cells were added that had

been transduced to express the MART-1$_{26-35}$/HLA-A2-specific TCR (see Supplementary Fig. 7). The T cell response was read out at day 6 or 7 after co-culture. **c**, **d** CD8+ T-cell proliferation to (**c**) MART-1$_{15-40}$ long peptide or (**d**) dead Mel526 cell debris based on CTV dilution. Upper panel, primary flow cytometric data. Lower panel, quantification of % MART-1$_{26-35}$/HLA-A2-specific (tetramer+) cells within CTV-negative (⁻) CD8+ T-cells. **e** CTL response to dead Mel526 cell debris based on intracellular Granzyme B staining. Upper panel, primary flow cytometric data. Lower panel, quantification of % Granzyme B+ cells among CD8+ T-cells. Data were pooled from four independent experiments ($n = 4$) in (**c**), six independent experiments ($n = 6$) in (**d**) and five independent experiments ($n = 5$) in (**e**), each experiment had technical duplicates. $p < 0.05^*$, $p < 0.01^{**}$, $p < 0.001^{***}$ (two-sided Mann–Whitney test). Data are shown as means ± standard error of the mean (SEM).

the clinically favorable DC_S3 signature, but had very few transcripts in common with the unfavorable DC signatures (Fig. 4d). All together, we conclude that tumor-infiltrating DC3[17], the mature DC in the IL-32[hi] TME[34] and the tumor-infiltrating DC_S3[18] are most likely derived from cDC1 and appear to reflect conditions of CD4[+] T-cell help.

### CD4[+] T-cell help signature in the TME is associated with CTL and Th1 cell infiltration and positive clinical outcome in cancer patients

The cognate interaction of CD4[+] T-cells and CD8[+] T-cells with cDC1 promotes the CTL response and Th1 differentiation in mice[15,29,35]. To test whether the same was true in human, we performed correlation analysis between DC signatures and different T-cell differentiation signatures[36] (Supplementary Data 2) in a skin cutaneous melanoma (SKCM) patient cohort listed in The Cancer Genome Atlas (TCGA). The DC3-, DC_S3- and cDC1 "help" signatures, as well as the 66 transcripts present in both cDC1 "help" signature and DC_S3 (Fig. 5a) correlated with activated CD8[+] T-cells ($R > 0.85$), effector/memory CD8[+] T-cells ($R > 0.92$), and CD4[+] Th1-cells ($R > 0.9$), but not with CD4[+] Th2-cells ($R < 0.65$) in the TME (Fig. 5b–e). The cDC1 "help" signature and the shared signature between "helped" cDC1 and DC_S3 had a higher degree of correlation with activated CD8[+] T-cells ($R = 0.93$, 0.92 respectively) (Fig. 5d, e) as compared to the DC3[17] and/DC_S3[18] signatures ($R = 0.85$, 0.89 respectively) (Fig. 5b, c). Moreover, the shared signature between "helped" cDC1 and DC_S3 (Fig. 5a) highly corelated with signatures of CD8[+] T-cells in state S3 and CD4[+] T-cells in state S1 that are associated with longer OS[18] (Supplementary Fig. 12b).

To further assess the clinical relevance of our findings, we performed Kaplan–Meier survival analysis using a melanoma TCGA cohort and another melanoma cohort that received PD-1 blockade[37]. All above-mentioned DC signatures significantly associated with longer OS in the TCGA melanoma cohort (Fig. 5f), with the lowest significance in case of the tumor-infiltrating DC3 signature ($HR = 0.63$; $p = 6.1 \times 10^{-5}$) and the highest significance in the shared signature between "helped" cDC1 and DC_S3 ($HR = 0.5$; $p = 5.2 \times 10^{-7}$) (Fig. 5f). Regarding PD-1-targeting immunotherapy[37], the tumor-infiltrating DC3 signature[17] did not predict responsiveness ($Z = -1.68$; $p = 0.09$), whereas the tumor-infiltrating DC_S3 signature[18] did ($Z = -2.36$; $p = 0.018$) (Fig. 5g). Strikingly, higher predictive values of responsiveness to PD-1 blockade were observed with the cDC1 "help" signature ($Z = -2.62$; $p = 0.0087$) and the shared signature between "helped" cDC1 and DC_S3 ($Z = -2.59$; $p = 0.0094$) (Fig. 5g).

The combined results indicate that the cDC1 "help" signature reflects CD4[+] T-cell help delivered to CD8[+] T cells in the TME. This scenario then apparently optimizes CTL and Th1 differentiation, which translates into a better T-cell mediated tumor control in a great variety of human solid tumor types.

## Discussion

Mouse studies have shown that DC-licensing by activated CD4[+] T-cells proceeds largely but not exclusively by CD40 engagement on the DC and upregulation of co-stimulatory ligands CD80, CD86, and CD70[7,28,29,38] and the specific cytokines IL-12 and IL-15[35] that promote CD8[+] T-cell clonal expansion, effector and memory differentiation[7]. At the same time, CD4[+] T-cells are instructed to proliferate and complete Th1 differentiation[39]. A recent mouse study pinpointed that CD40 activation on cDC1 is essential for priming a CD4[+] T-cell response to cell-associated antigen and help for CTL-based tumor rejection[16]. A critical role of cDC1 in tumor control was previously shown in the mouse[40] and suggested for human by the correlation of cDC1 transcript abundance with OS[13].

We here identify the transcriptomic imprint of DC licensing by CD4[+] T-cells in human cancer and pinpoint the cDC1 as the recipient of CD4[+] T-cell help in the human TME. In accordance with mouse studies, the gene expression signature in "helped" human cDC1 correlated with

CTL and Th1 infiltration in the tumor and improved tumor control. Among ten transcriptomically defined cellular ecosystems in thousands of human cancers, three positively correlated with OS[18]. In two of these, CE9 and CE10, conventional CD4[+] and CD8[+] T-cells and DC coexist, in agreement with a scenario wherein CD4[+] T-cell help for the CTL response can be delivered[18]. In CE9, the DC_S3 state was present that we demonstrated to be greatly enriched in the cDC1 "help" gene expression signature. In this way, we connected our in vitro generated cDC1 "help" signature to the physiological situation in human cancer. Importantly, we showed that the DC_S3 state almost exclusively had features of "helped" cDC1 and not of either "helped" or "non-helped" cDC2, pDC or moDC.

Thus far, the scenario of CD4[+] T-cell for the CTL response has been described to occur in secondary lymphoid organs. However, during the effector phase in non-lymphoid (e.g. tumor) tissues, T-cells also interact with DC. DC can be recruited to the tumor, or be generated within the tumor tissue from progenitors[41]. XCL1 produced by NK- and T-cells[40,42] plays an important role in recruiting XCR1-expressing cDC1[43]. cDC1 locally support T-cell recruitment and further effector differentiation and proved to be the superior myeloid cell type in stimulating CD8[+] T-cells. cDC1 may also regulate innate immunity in TME[44]. Finding the cDC1 "help" signature in the TME argues that CD4[+] T-cell help is delivered to cDC1 in the TME and emphasizes the importance of T cell-DC crosstalk in this environment.

In this paper, we define the imprint of CD4+ T-cell help in human cDC1 and other DC types at the molecular level. The molecular basis of DC licensing has not been described before in either human or mouse. The cDC1 and to a lesser extent the cDC2 were responders to CD4[+] T-cell help, while pDC and moDC were inert. The greatest transcriptomic change occurred in cDC1 that showed increased expression of many molecules that have been implicated in T-cell priming or can easily be inferred to play such a role. Many features important for T-cell priming were confirmed at the protein level. We found that the CD4[+] T-cell help response was more explicit and sometimes unique in these features than a response of cDC1 to combined PRR stimulation under our in vitro conditions that evidently do not incorporate all in vivo variables. Upregulation of ubiquitination, proteasome subunits, TAP1/ 2 and MHC class I molecules indicates enhanced protein processing into presentable peptides and enhanced antigen presentation by MHC class I, congruent with more efficient CD8[+] T-cell priming[45,46]. This feature was specifically stimulated by CD4[+] T-cell help as compared to PRR stimulation. Upregulation of CD80/CD86 and CD70 promote both CD4[+] and CD8[+] T-cell costimulation via their respective receptors CD28 and CD27 that in concert with IL-15 promote cell division, survival, CTL and Th1 effector differentiation and CTL memory differentiation[7]. In agreement with mouse studies[15,28,29], we demonstrated the importance of CD4[+] T-cell mediated CD70 induction on the human cDC1 for CTL cross-priming by antibody intervention. The observed upregulation of PD-L1 is not counteractive to T-cell priming, since CD80 can heterodimerize with PD-L1 and form a costimulatory ligand for CD28 that is resistant to negative control by PD-1 and CTLA4[47].

Acquiring and maintaining chemokine and chemokine receptor expression by DC in TME is important for better tumor control[12]. The upregulation of CCR7 on "helped" cDC1 is intriguing, since cDC1 depend on CCR7 to migrate from the TME into draining lymph nodes for tumor-specific T-cell priming in mouse models[48] and the tumor-infiltrating DC3[17] and DC_S3[18] that are found in the TME of diverse cancers express CCR7. Moreover, LAMP3[+] DC that are comprised in the DC3[17] showed a trajectory from TME towards tumor draining lymph nodes according to transcriptome data[31]. Furthermore, "helped" cDC1 upregulate CXCL9/10/11 that may promote attraction of CXCR3[+] effector T-cells into the TME, as shown for tumor-infiltrating cDC1[49,50]. So, interaction between activated CD4[+] T-cells and cDC1 in TME may initiate a positive loop between TME and tumor-draining lymph nodes

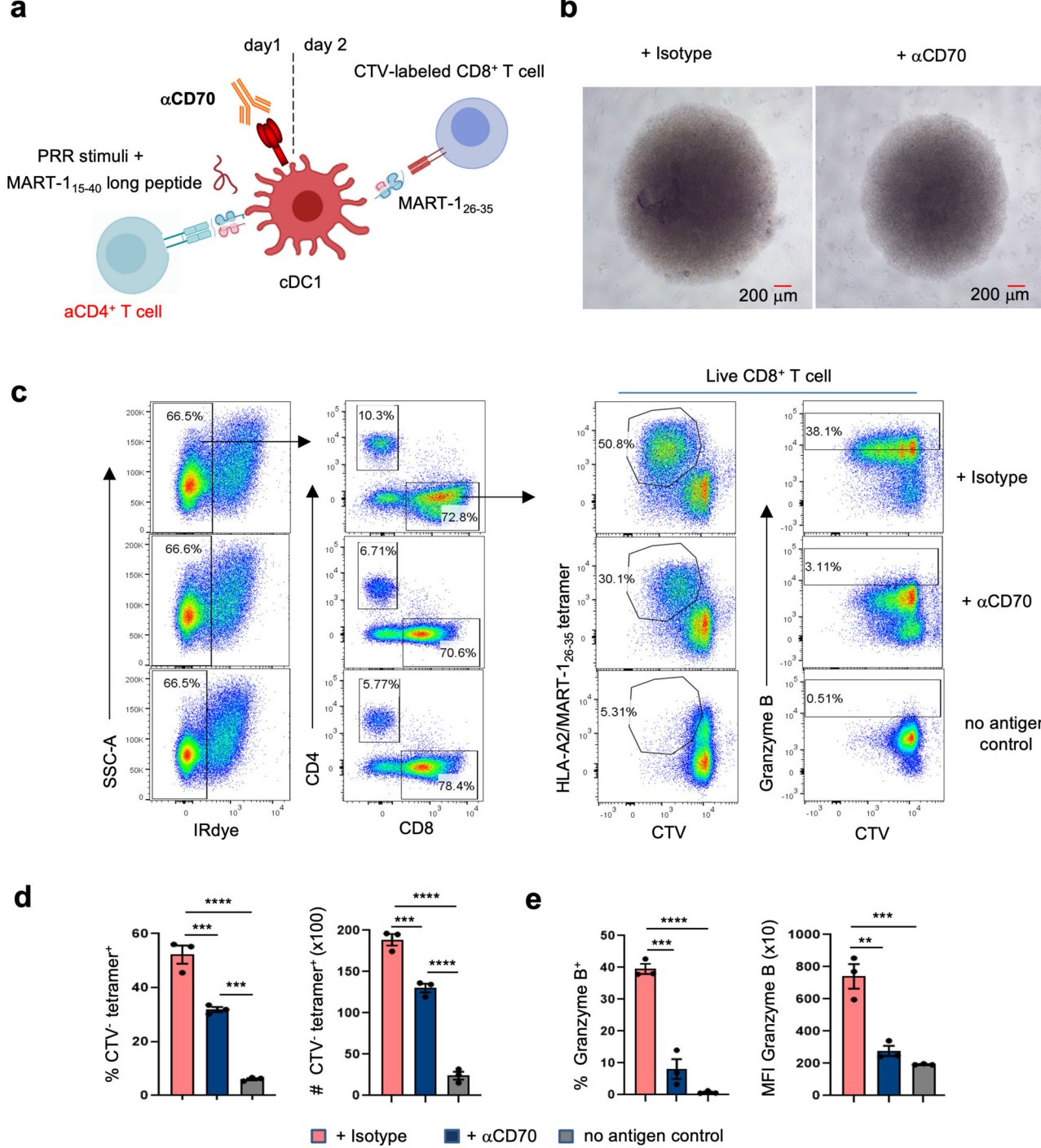

**Fig. 3 | CD70 on human cDC1 promotes the CTL response to CD4⁺ T-cell help.**
Sorted cDC1 were incubated with CD70 blocking antibody (CLB-2F2[54]) or isotype control and entered in the priming system with MART-1[15-40] long peptide in presence of activated (a) CD4⁺ T-cells. **a** Schematic depiction of the tumor antigen-specific CTL priming system in the presence of CD70 blockade. **b** Light microscopic images depicting CTL priming system at day 7 under indicated conditions. Images are representative of three independent experiments. **c** Flow cytometry plots depicting MART-1-specific CD8⁺ T-cell proliferation based on CTV dilution and CTL differentiation based on intracellular Granzyme B staining in response to antigen presented by cDC1. **d** Quantification of the % MART-1[26-35]/HLA-A2-specific (tetramer⁺) cells within CTV⁻negative (⁻) CD8⁺ T-cells (left) and the number (#) of live MART-1-specific CTV⁻CD8⁺ T-cells (right). **e** Quantification of the % Granzyme B⁺ cells among CD8⁺ T-cells (left) and MFI of Granzyme B expressed by CD8⁺ T-cells (right). **d**, **e** share the same legends. Data are pooled from three independent experiments ($n = 3$) in (**d**, **e**), each with technical duplicates. $p < 0.05^*$, $p < 0.01^{**}$, $p < 0.001^{***}$ (One way ANOVA). Data are shown as means ± SEM in (**d**, **e**).

with sustained cDC1 migration and recruitment of effector T-cells and anti-tumor immunity.

"Helped" cDC1 also upregulated IL-32 that was originally shown to promote generation of moDC[51] and IL-12 and IL-6 production[52] by DC. An IL-32[hi] TME was strongly enriched, not only in melanoma, but in all available TCGA cohorts, for a previously described maturing DC signature[33]. In melanoma patients, an IL-32[hi] TME correlated with presence of mature DC, M1 macrophages and CD8⁺ T-cells, better OS and response to PD-1 checkpoint blockade. IL-32 injection in mouse tumors supported a causal link between IL-32, tumor-infiltrating DC/

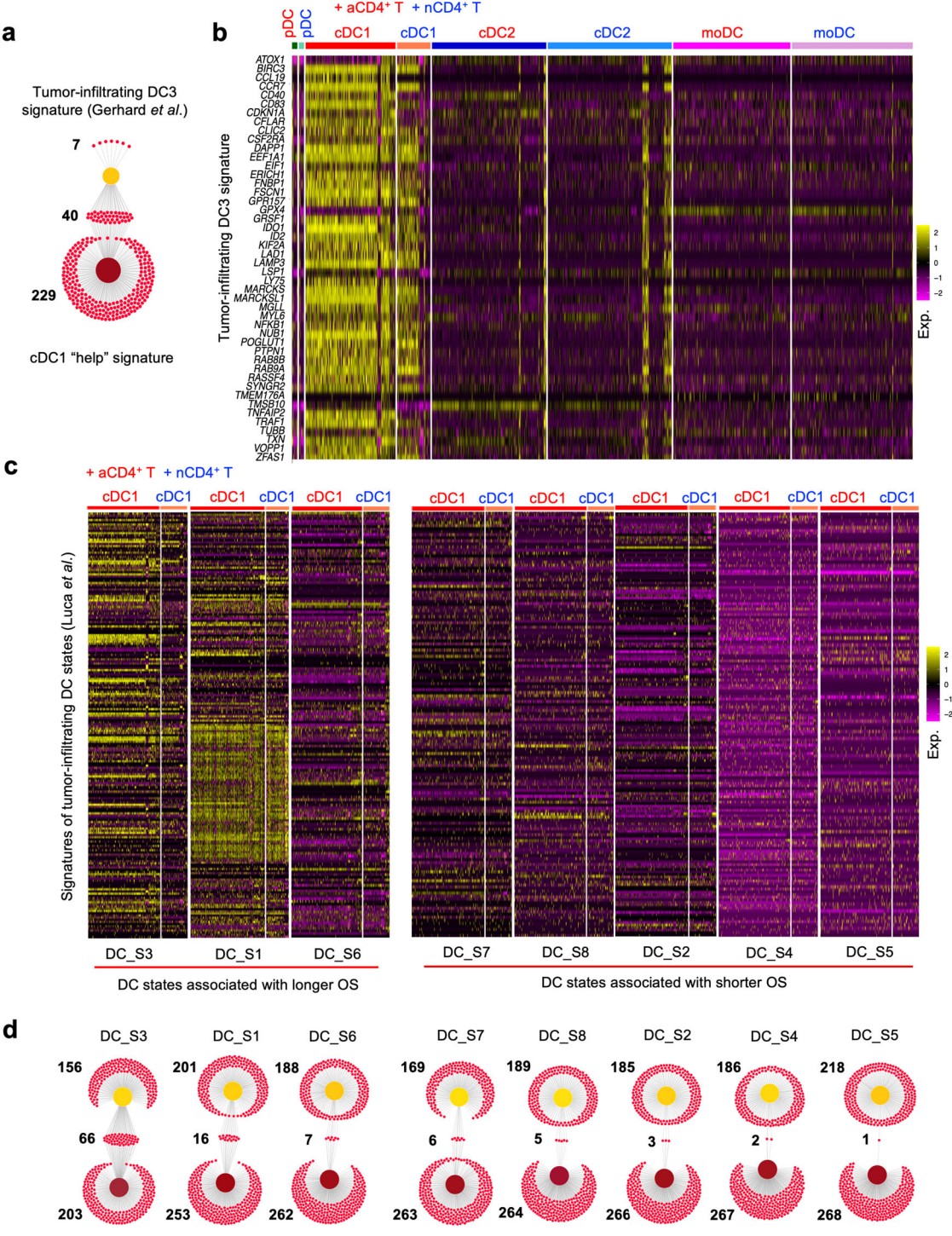

**Fig. 4 | The "helped" cDC1 state is transcriptionally related to clinically favorable tumor-infiltrating DC_S3 state.** To investigate the relationship between "helped" cDC1 and tumor-infiltrating DC that are conserved across multiple human solid tumor types, the cDC1 "help" signature was cross-compared with a tumor-infiltrating DC3 signature from Gerhard et al.[17] and with tumor-infiltrating DC signatures of 8 different states that are associated with either longer or shorter overall patient survival from Luca et al.[18]. **a** Venn diagram depicting number of overlapping genes between the cDC1 "help" signature and the tumor-infiltrating DC3 signature. **b** Heatmaps depicting the expression of tumor-infiltrating DC3 signature genes[17] in each DC subset under "help" (aCD4+ T) or "no help" (nCD4+ T) conditions. **c** Heatmap depicting the expression of signature genes from the indicated tumor-infiltrating DC states (DC_S3 etc.) that are associated with longer or shorter overall survival (OS)[18] in cDC1 under "help" (aCD4+ T) or "no help" (nCD4+ T) conditions. **d** Venn diagrams depicting numbers of overlapping genes between the cDC1 "help" signature and the tumor-infiltrating DC signatures of 8 different states defined in the study of Luca et al.[18].

macrophage activation and CTL recruitment to the tumor[34]. The cDC1 "help" signature comprised half of the mature DC signature in the IL-32[hi] TME, suggesting that IL-32 may promote cDC1 maturation in an autocrine manner.

Our collective data indicate that contact with activated CD4[+] T-cells programs cDC1 to boost all molecular programs that are important for optimal anti-tumor immunity. Our data argue that the cDC1 "help" signature can serve as prognostic and predictive biomarker for cancer patients. Furthermore, they underline the importance of cDC1 and CD4[+] T-cell help for effective CTL-based anti-tumor immunity. This knowledge is important for the design of DC vaccination strategies that should center on cDC1 and ensure that these cells have the functional properties endowed by CD4[+] T-cell help. In one vaccine trial, in vitro generated moDC were transfected to express TLR4, CD70 and CD40 ligand to improve their efficacy[53], but we now show that "helped"/licensed cDC1 have a plethora of optimized functions that explain their potency as compared to moDC.

## Methods

### Human peripheral blood samples
Human peripheral blood mononuclear cells (PBMCs) were obtained in accordance with the Declaration of Helsinki and the Dutch rules with respect to the use of human materials from volunteer donors. Buffy coats from healthy anonymized donors were obtained after their written informed consent, as approved by Sanquin's internal ethical board. PBMCs were isolated from buffy coats using Ficoll-Paque Plus density gradient centrifugation (GE Healthcare) and cells were cryopreserved till further use. DC were isolated from HLA-A2[+] donors, while CD4[+] and CD8[+] T cells used in this study were used regardless of their HLA type and were not necessarily from the same donor.

### Fluorescence-activated cell sorting (FACS)
For in vitro DC-T cell co-culture, bulk mRNA-Seq and CITE-Seq experiments, PBMCs were directly used for FACS. For ex vivo DC-CD4[+] T-cell co-culture experiments, CD19[+] cells were depleted before sorting using CD19 magnetic MicroBeads (MACS), according to the manufacturer's protocol. Staining was performed at 4 °C for 45 min in flow cytometry staining buffer (BD Biosciences). The following antibodies were used: from BioLegend: CD1c (clone L161), CD3 (clone OKT3), CD4 (clone OKT4), CD8 (clone SK1), CD11c (clone Bu15/3.9), CD14 (clone M5E2), CD19 (clone HIB19), CD25 (BC96), CD45RA (clone HI100), CD141 (clone M80), CD206 (clone 15-2), CD303 (clone 201 A), HLA-DR (clone L243); from BD Biosciences: HLA-DR (clone G46-6); from Miltenyi Biotec: CD141 (clone REA674). Near-IR Dead Cell Stain Kit (Invitrogen), Zombie Red Fixable Viability Kit (BioLegend) or 7-amino-actinomycin D (7-AAD, eBioscience) were used to exclude dead cells. In each sorting, cDC2 (within the same sample) was used as a negative control for gating ex vivo moDC. Detailed information regarding these antibodies can be found in the Reporting Summary. In order to prevent clump formation from dead cells, 0.01% DNase (Invitrogen) was added before sorting. Cell sorting was performed on BD FACSAria[TM] Fusion or BD FACSAria III (BD Biosciences).

### Flow cytometry
Cell surface staining: Staining was performed at 4 °C for 30 min in flow cytometry staining buffer (BD Biosciences). The following antibodies (Supplementary Data 3) were used: from BioLegend: CD1a (clone HI149), CD1c (clone L161), CD3 (clone OKT3), CD4 (clone OKT4), CD8 (clone SK1), CD11c (clone Bu15/3.9), CD14 (clone 63D3), CD28 (clone CD28.2), CD40 (clone 5C3), CD40L (clone SA047C3), CD44 (clone C44Mab-5), CD45RA (clone HI100), CD62L (clone Dreg-56), CD69 (clone FN50), CD70 (clone 113-16), CD80 (clone 2D10), CD83 (clone HB15e), CD86 (clone IT2.2), CD95 (clone Dx2), CD137 (clone 4B4-1), CD141 (clone M80), CD206 (clone 15-2), CD209 (clone 9E9A8), CD303 (clone 201A), CCR7 (clone G043H7), CXCR3 (clone G025H7), HLA-A2

(clone BB7.2), HLA-ABC (clone W6/32), HLA-DR (clone L243), PD-L1 (clone 29E.2A3); from BD Biosciences: CD14 (clone M5E2), CD27 (clone L128), CD45RO (clone UCHL1), HLA-DR (clone G46-6); from Miltenyi Biotec: CD141 (clone REA674); from ImmunoTools: CD8 (clone HIT8a). APC-conjugated HLA-A2/MART-1[26-35] tetramers were added together with cell surface staining antibodies. Near-IR Dead Cell Stain Kit (1:1000, Invitrogen), Zombie Red Fixable Viability Kit (1:800, BioLegend) or 7-Aminoactinomycin D (1:20 7-AAD) were used to discriminate between live and dead cells.

For intracellular staining, protein transport inhibitor (BD Golgi-Plug) (1:1000) was added into the culture for 3 h before cells were harvested and analyzed by flow cytometry. After surface staining, cells were fixed and permeabilized using the BD Cytofix/Cytoperm kit (BD Biosciences), according to the manufacturer's protocol. The following antibodies were used: from BioLegend: β2m- (clone A17082A), Granzyme B (clone QA16A02), CXCL9 (clone J1015E10), CXCL10 (clone J034D6), IFNγ (clone B27), TNFα (clone MAb11); from Cell Signaling Technology: cleaved caspase 3 (clone D1751); from Abcam: mouse anti-human Melan A (clone A103, 1:200); from Bioss: rabbit anti-human TAP1 and TAP2 polyclonal antibodies; from Thermo Fisher Scientific: goat anti-rabbit IgG(H + L) Alexa Fluor 488 (1:200) and goat anti-mouse IgG(H + L) Alexa Fluor 647 (1:300) secondary antibodies. Specific stainings were confirmed by goat IgG Alexa Fluor 647 Isotype (BioLegend) or Fluorescence Minus One (FMO) control. Antibody stocks were diluted 1:50 for use unless stated otherwise. Detailed information regarding these antibodies can be found in the Reporting Summary. Flow cytometry was performed using a BD LSR Fortessa[TM], BD FACS-Symphony[TM] A5 SORP flow cytometer (BD Biosciences) or Cytek Aurora spectral flow cytometer. Data were analysed using FlowJo[TM] software version 10.7.2 (BD Biosciences).

### Bulk mRNA sequencing of human ex vivo DC subsets
PBMCs from 3 healthy donors were stained with the appropriate antibodies (Supplementary Fig. 1a) and within the live CD3[-]CD19[-]HLA-DR[+] population, pDC (CD11c[-]CD14[-]CD303[+]), cDC1 (CD11c[+]CD14[-]CD141[+]), cDC2 (CD11c[+]CD14[-]CD1c[+]), and moDC (CD11c[+]CD14[+]CD1c[+]CD206[+]) were sorted. Then cells were washed in ice-cold PBS and resuspend in buffer RLT (Qiagen). Total RNA isolation was performed according to manufacturer's protocol using the RNeasy MinElute Cleanup Kit (Qiagen). Quality and quantity of the total RNA were assessed on a 2100 Bioanalyzer using a Nano chip (Agilent). Only RNA samples having an RNA Integrity Number (RIN) > 8 were subjected to library generation. Strand-specific cDNA libraries were generated using the TruSeq Stranded mRNA sample preparation kit (Illumina) according to the manufacturer's protocol. The libraries were analyzed for size and quantity of complementary DNAs (cDNAs) on a 2100 Bioanalyzer using a 7500 chip (Agilent), diluted and pooled in equimolar ratios into a multiplex sequencing pool. The libraries were sequenced as 65 base single reads on a HiSeq2500 with V4 chemistry (Illumina).

### Bulk mRNA-seq analysis of human ex vivo DC subsets
Bulk RNA-seq reads were aligned to the human reference genome (Homo sapiens GRCh37.66.) using TopHat software (version 2.1.0). Only unique mapped reads were used for gene expression analysis. Read counts were normalized using Limma (Version 3.22.7) and count per million (CPM) were calculated. Differential-expression analysis and Hierarchical clustering were performed using Limma (Version 3.22.7). Differential mRNA expression was considered significant with p-value <0.01 and log$_2$-fold change (FC) >2 or <−2. Heatmap of 114 significantly differentially expressed genes (DEGs) between DC subsets across 374 CD markers from Human Protein Atlas was generated based on log$_2$FC using Qlucore Omics Explorer (version 3.7). Gene ontology biological process analysis was performed using Ingenuity Pathway Analysis software (IPA, Qiagen) to identify pathways that were differentially

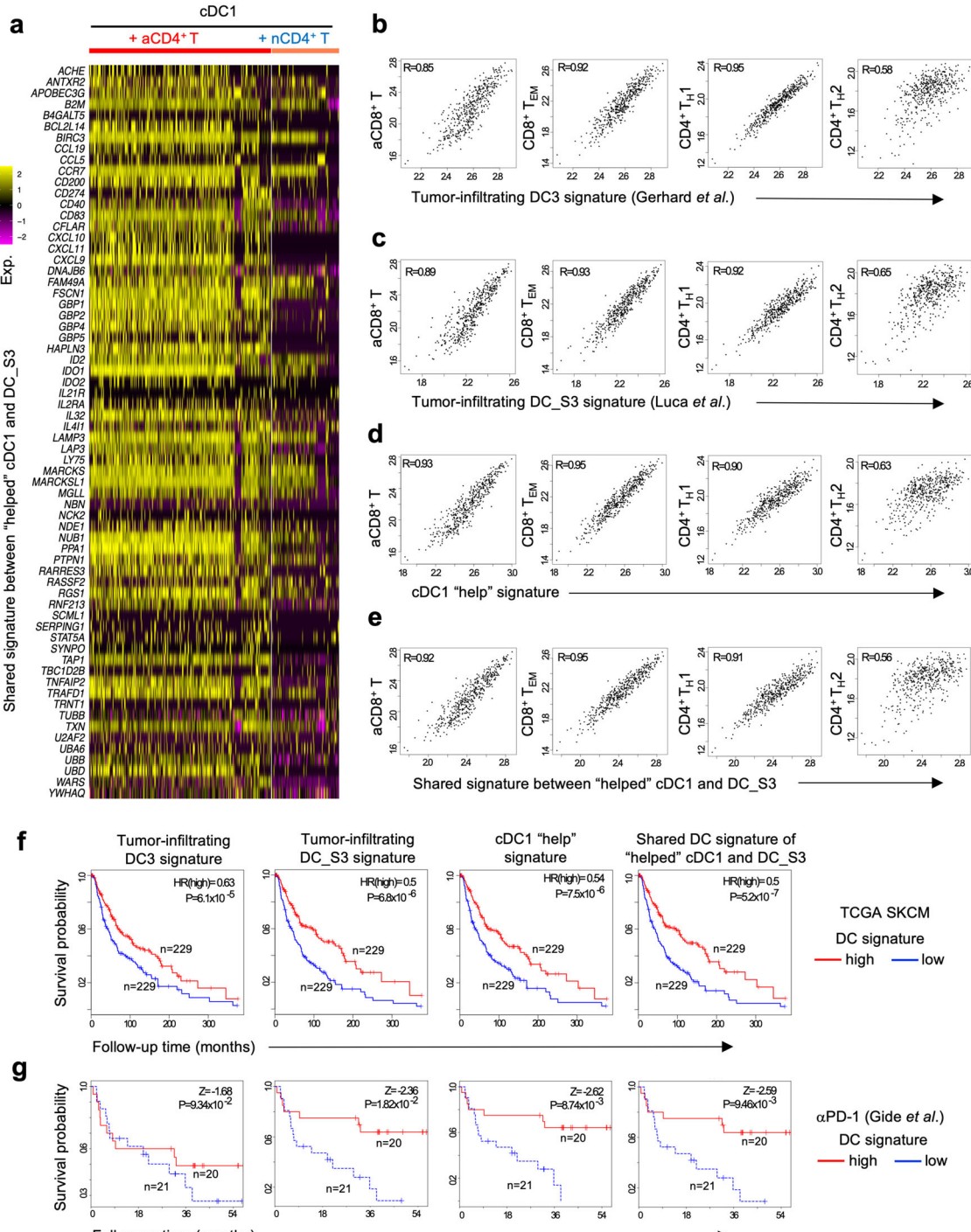

**Fig. 5 | "Helped" cDC1 state positively predicts clinical outcome in cancer patients. a** Heatmap depicting the expression of shared genes between cDC1 "help" signature and tumor-infiltrating DC_S3 signature (66 genes) in cDC1 under "help" (aCD4+ T) or "no help" (nCD4+ T) conditions. **b–e** Pearson's correlations between (**b**) tumor-infiltrating DC3 signature[17], (**c**) tumor-infiltrating DC_S3 signature[18], (**d**) cDC1 "help" signature or (**e**) shared signature between "helped" cDC1 and tumor-infiltrating DC_S3 and defined tumor infiltrating T-cell signatures denoting activated (**a**) and effector memory (EM) CD8+ T-cells and Th1 and Th2 CD4+ T-cells[36] within TCGA SKCM dataset (*n* = 458). R, correlation

coefficient. **f, g** Kaplan–Meier curves revealing prognostic/predictive value of tumor-infiltrating DC3 signature[17], tumor-infiltrating DC_S3 signature[18], cDC1 "help" signature or shared signature between "helped" cDC1 and tumor-infiltrating DC_S3 for (**f**) melanoma patient's OS in TCGA SKCM cohort (*n* = 458; baseline transcriptome), or (**g**) response to anti-PD-1 immunotherapy[37] (*n* = 41; baseline transcriptome). High or low metagene expression subgroups of patients were based on a median expression cut-off. *p*-value was calculated using Log-rank test/Mantel-Cox test (TCGA SKCM cohort) or CoxPh hazard ratios (HR), depicted as Z-scores in the anti-PD1 immunotherapy trial (*p* < 0.05 is considered significant).

expressed between DC subsets based on -log$_{10}$(adjusted p-value) of DEGs between DC subsets.

## Retroviral transduction of CD8$^+$ T cells with MART-1$_{26-35}$/HLA-A2-specific TCR

This method was adapted from a described protocol[25] None-tissue culture treated 24-well plates (BD Falcon) were coated with 10 µg/ml RetroNectin (Takara) at 4 °C for 24 h, blocked with 2% Bovine Serum Albumin (BSA, Sigma) for 30 min at room temperature (RT), then washed with Phosphate Buffered Saline (PBS) twice. CD8$^+$ T-cells were cultured in RPMI-1640 (Gibco, Life Technologies), supplemented with 10% FCS (Sigma), penicillin/streptomycin (Sigma), β-mercaptoethanol (Sigma), in the presence of human (h)IL-2, hIL-7 and hIL-15 (Miltenyi Biotec) each at 10 ng/ml and human T-Expander CD3/CD28 Dynabeads (2 cells:1 bead, Thermo Fisher Scientific) for 2–3 days before transduction. For transduction, CD8$^+$ T-cells with beads attached were spun down and resuspended in retrovirus-containing medium from packaging cells supplemented with 10 ng/ml hIL-2/hIL-7/hIL-15 (Miltenyi Biotec) and plated 0.5 × 10$^6$ cells per well. Plates were centrifuged at 800 g for 90 min at RT in a table-top centrifuge with ascending speed of 3, descending speed of 0. Cells were cultured for 24 h, next virus-containing supernatant was removed, and cells were expanded in medium with the cytokine cocktail and CD3/CD28 Dynabeads for 7 days. Next, the beads were removed and cells were rested in medium with the cytokine cocktail for 3 days before use in in vitro CTL priming experiments.

## Tumor antigen-specific CTL priming platform

To test the impact of CD4$^+$ T-cell help on CTL priming, either activated- or naïve conventional CD4$^+$ T-cells were used. Naïve CD4$^+$ T-cells were flow cytometrically sorted on a CD3$^+$HLA-DR$^-$CD4$^+$CD25$^{-/low}$CD45RA$^+$ phenotype and cultured for 2–3 days at 0.5 × 10$^6$ cells/well in 96-well round bottom plates (BD Falcon) in medium with cytokines as indicated above, in absence (naïve cells) or presence (activated cells) of monoclonal antibodies against CD3 (clone CLB-T3/4.E, IgE isotype, Sanquin, 0.1 µg/ml) and CD28 (clone CLB-CD28/1, Sanquin, 0.2 µg/ml), before being added into the CTL priming culture. To create conditions of antigen crosspresentation, MART-1$_{15-40}$ long peptide (KGHGHSYT-TAEELAGIGILTV), dead melanoma cells (Mel526, or incidentally Mel-AAT as indicated) were used. The Mel526 cell line originates from the S.A. Rosenberg laboratory (National Institutes of Health, Bethesda, USA). The MelAAT cell line was derived from a melanoma patient at The Netherlands Cancer Institute. To induce apoptotic cell death, melanoma cells were treated with 100 ng/ml tumor necrosis factor-related apoptosis-inducing ligand (TRAIL, Merck) and 10 ng/ml Fas Ligand (FASL, AdipoGen) for 3 days, then cell was collected and pelleted by centrifugation.

Flow cytometrically sorted ex vivo HLA-A2$^+$ DC were treated with either activated- or naive CD4$^+$ T-cells in 1:1 ratio for 2 h in Iscove's Modified Dulbecco's Medium (IMDM) (Gibco, Life Technologies), supplemented with 1% FCS. Then MART-1$_{15-40}$ long peptide (20 µg/ml) or dead melanoma cells, and PRR stimuli (LPS 50 ng/ml + poly I:C 20 µg/ml + R848 3 µg/ml, InvivoGen) were added. Anti-CD70 mab 2F2[54] or IgG1 isotype (Invitrogen) at a concentration of 5 µg/ml were additionally added where indicated. After 12–16 h, cell supernatant was washed away. Then MART-1$_{26-35}$/HLA-A2-specific TCR transduced CD8$^+$ T-cells were added into the culture in 1 DC: 5–10 CD8$^+$ T-cell ratio and cultured for 6–7 days in RPMI-1640, supplemented with 10% FCS and 0.2 ng/ml hIL-2/hIL-7/hIL-15. The donors of CD4$^+$ and CD8$^+$ T-cells were used without regard to HLA-A2 positivity. To trace proliferation, CD8$^+$ T-cells were labeled with CTV before being added into the CTL priming platform. 50 ng/ml Phorbol 12-myristate 13-acetate (PMA, Sigma), 1 µg/ml Ionomycin (Thermo Fisher Scientific) and Protein transport inhibitor (BD GolgiPlug, BD Biosciences) (1:1000) were

added into the culture for 2 h before cells were harvested and analyzed by flow cytometry.

## Hashtag Single cell (sc)RNA-seq

pDC, cDC1, cDC2 and moDC were flow cytometrically sorted as outlined before from PBMCs of three independent healthy donors and cultured with activated- or naïve CD4$^+$ T-cells prepared as outlined before at a ratio of 10,000 DCs with 4000 CD4$^+$ T-cells per well in 96-well round-bottom plates overnight. Then cells in each DC- and CD4$^+$ T-cell help condition were labeled with 8 distinct hashtag oligonucleotides (HTO)-conjugated antibodies to β2m and CD3 antibody derived tag before pooling all the samples together. The HTO labeling thus allowed pooled analysis of all samples to avoid batch effects and for improve detection of multiplets. Then cells were encapsulated using the 10X Chromium single cell 3' v2 chemistry kit according to the manufacturer's instructions. Libraries were prepared as previously described with minor modifications[24]. In brief, amplification of cDNA was performed in the presence of 2 pM of an antibody-oligo-specific primer to increase the yield of antibody-derived tags (ADTs). The amplified cDNA was then separated by SPRI selection (Beckman Coulter Life Science) into cDNA fractions containing messenger-RNA-derived cDNAs (larger than 300 base pairs) and ADT-derived cDNAs (smaller than 180 base pairs), which were further purified by additional rounds of SPRI selection. Independent sequencing libraries were generated from the mRNA and ADT cDNA fractions, which were quantified, pooled and sequenced together on an Illumina Nextseq to a depth of 1300 million reads per gene expression library and 20 million reads per ADT library. Of 25,000 sequenced cells with average 239,576 reads per cell, 5455 cells passed quality control with a median of 3866 genes detected per cell.

## scRNA-seq data generation and processing

(1) Quality Control: The count matrix, ADT and HTO matrix obtained from the sequencing data of 25,000 cells processed with the Cell Ranger 2.2.0 software was loaded in *Seurat* v3.6.1 in R4.0. We removed all cells that expressed <500 genes or that had >10% and >40% of their transcripts mapped to mitochondrial and ribosomal genes, respectively. The multiplets and negatives were identified and removed using the *HTODemux* function in *Seurat*. Transcriptomes from CD4$^+$ T-cells were excluded based on *CD3D* mRNA expression and HTO against CD3 (*CD3D* mRNA expression <=0.1 and ADT expression <200) to further analyze data from DCs. *Preprocessing*: After removing unwanted cells from the dataset, a global-scaling normalization method "LogNormalize" was employed that normalizes the feature expression measurements for each cell by the total expression, multiplies this by a scale factor (10,000 by default), and log-transforms the result. In the next step variable features that exhibit high cell-to-cell variation in the dataset were identified using *FindVariableFeatures* function. The scaling of the variable features was performed prior to dimensional reduction using PCA. The clusters were identified using the *FindCluster* function in Seurat that implements Louvain clustering method by default to identify the clusters. Finally, the non-linear dimensionality technique i.e. tSNE was used to visualize the cells in 2-dimensional space. (2) Supervised classification of single cells to cell types: The HTO information was mapped onto the clusters of the cells. At this step, a few cells with discrepant clustering and HTO tags were filtered out e.g. The clusters with cells from multiple tags or the cells from minority tags were removed. 2232 DC were finally obtained, which were further used for the preprocessing steps as explained above. Finally, the cell types are assigned to the cells based on the HTO tags. (3) Differential expression analysis: To identify the DEGs, *FindAllMarkers* function in Seurat was used. Wilcoxon Rank Sum test was used to identify the features specific to each cell type. The genes that are detected in 40% of the cells of the cluster and show log$_2$FC > 0.5 between two groups of cells

were used for testing. The genes with adjusted *P*-value < 0.05 were considered significant DEGs.

### Gene ontology (GO) analysis and gene set enrichment analysis (GSEA)

Log$_2$FC of the 577 DEGs between cDC1 cultured with activated- or naïve CD4$^+$ T-cells identified by scRNA-Seq were used for GO analysis and GSEA. For GO analysis, IPA software was used. For GSEA, the GSEA software (version 4.1.0) (http://broadinstitute.org/gsea) and Reactome pathway database (https://www.gsea-msigdb.org/gsea/msigdb/index.jsp) were employed with default parameters to calculate the enrichment and create GSEA plots.

### Comparative, correlation, survival analysis using public datasets

Comparative analysis with tumor-infiltrating DC signatures: Tumor-infiltrating DC3 signature[17], mature DC signature in IL-32$^{hi}$ TME[33,34] and tumor-infiltrating DC_S3 signature[18] were acquired through these publications. Next, cDC1 "help" signature was compared with these DC signatures (Supplementary Data 1). The scaled expressions of overlapped genes between cDC1 "help" signature and two tumor-infiltrating DC gene signatures were depicted as heatmaps.

*Correlation analysis in TCGA melanoma datasets* was carried out between tumor-infiltrating DC3 signature[17] tumor-infiltrating DC_S3 signature[18], cDC1 "help" signature or cDC1 "help" signature presented in tumor-infiltrating DC_S3 and various T-cell gene signatures[18,36] for The Cancer Genome Atlas' (TCGA) skin cutaneous melanoma (SKCM) patients' dataset (*n* = 458 patients) or combined datasets from breast invasive carcinoma (BRCA), colon adenocarcinoma (COAD), liver hepatocellular carcinoma (LIHC), lung adenocarcinoma (LUAD), ovarian serous cystadenocarcinoma (OV) and rectum adenocarcinoma (READ) patients (*n* = 2702) using the GEPIA2 computational work-flow[55] based on the UCSC Xena project (http://xena.ucsc.edu). The T-cell gene signatures consisted of activated CD8$^+$ T-cells, effector-memory CD8$^+$ T-cells, Type-1 polarized CD4$^+$ T-cells and Type-2 polarized CD4$^+$ T-cells, which were derived from an existing pan-cancer immunogenomic resource[36], as well as CD8$^+$ T-cell and CD4$^+$ T-cell signatures from CE9[18] (Supplementary Data 2). These signatures were used to carry-out pair-wise gene expression correlation analysis using Pearson's correlation method (non-log scale values were used for calculation, but log-scale axis was used for visualization).

*Survival analysis in the TCGA melanoma datasets* was carried out using tumor-infiltrating DC3 signature[17], tumor-infiltrating DC_S3 signature[18], cDC1 "help" signature or cDC1 "help" signature presented in tumor-infiltrating DC_S3. We accessed patient OS durations and tumor gene expression profiles for SKCM patients' dataset (*n* = 458), or combined datasets from BRCA, COAD, LIHC, LUAD, OV and READ patients (*n* = 2702) using the GEPIA2 computational work-flow[55], based on the UCSC Xena project (http://xena.ucsc.edu). Briefly, OS analysis was based on log-rank hypothesis test (the Mantel–Cox statistical test) that also estimated the Cox-proportional hazard ratio (HR) and the 95% confidence intervals accompanied by a Kaplan–Meier (KM) plot. Herein, expression threshold cut-off at median signature-expression level was used for splitting the patients into high-expression and low-expression sub-cohorts.

Survival analysis in immuno-oncology clinical trials was carried out using tumor-infiltrating DC3 signature[17], tumor-infiltrating DC_S3 signature[18], cDC1 "help" signature or cDC1 "help" signature presented in tumor-infiltrating DC_S3. These data and subsequent OS estimates were accessed using the 'Biomarker Evaluation' pipeline within a standardized TIDE computational workflow[56]. Herein, the prognostic effects were calculated as z-score deduced using the Coxph statistical model. These data were represented as Kaplan–Meier curves (at median expression cut-off). We accessed tumor transcriptomic data from melanoma patients (at pre-treatment/baseline) profiled before α-PD1 immunotherapy alone[37] (*n* = 41).

### Statistical analysis

Data, excluding those describing mRNA sequencing and comparative, correlation, survival analysis using public datasets, were analyzed using GraphPad Prism, Mann–Whitney test or one-Way ANOVA was used to determine significant differences between samples. Data are represented as means ± SD or means ± SEM. *P* value < 0.05 was considered statistically significant.

### Reporting summary

Further information on research design is available in the Nature Portfolio Reporting Summary linked to this article.

## Data availability

The bulk mRNA sequencing data utilized in Supplementary Fig. 1, 4 during the course of this study have been deposited in the GEO database with the accession number GSE218719. The cDC1 "help" signature has been listed in the Supplementary Data 1. The processed flow cytometry data are provided in the source data file. The scRNA sequencing data, other primary data and materials that support the findings of this study are available from the corresponding author upon reasonable request. Previously published tumor-infiltrating DC signatures can be accessed via DOI:10.1084/jem.20200264; 10.1172/jci.insight.138772 and 10.1016/j.cell.2021.09.014. Reactome database can be accessed via https://reactome.org/; TCGA cohorts used in survival analysis can be accessed via http://gepia2.cancer-pku.cn/#index. Source data are provided with this paper.

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

## Acknowledgements

This collaboration project was co-funded by the PPP Allowance to J.B., Y.X. and H.v.E. made available by Health~Holland, Top Sector Life Sciences & Health, to stimulate public-private partnerships. It was additionally funded by Oncode to J.B., Dutch Cancer Society (KWF) grant 11079 to J.B. and Y.X., Excellence of Science/EOS grant 30837538 of the Research Foundation Flanders (FWO) for the 'DECODE' consortium to J.B. and A.G. and an Antoni van Leeuwenhoek Foundation grant from the Bakker family to J.B. and Y.X. We thank all personnel of the flow cytometry facilities at the Netherlands Cancer Institute and Leiden University Medical Center for their excellent technique support, S. de Kivit, M. Mensink and E. Schrama from our own research group for providing primary cells. We thank T.N.M. Schumacher for providing the construct encoding the MART-1-specific TCR and J.J.M. van Dongen for financially supporting I.K. Images for clipart used figures are derived from BioRender.com.

## Author contributions

Y.X. designed the research, designed and performed experiments, analyzed/interpreted data and wrote the manuscript; J.B. designed the research, interpreted data and wrote the manuscript. X.L. designed and performed experiments, analyzed data and wrote the manuscript. I.K. contributed to scRNA-seq data analysis and helped writing the manuscript and making the figures. T.d.W. contributed to cell sorting and co-culture experiments and helped writing the manuscript, I.d.R., M.N. and R.K. contributed to the scRNA-seq experiment and helped in data analysis. A.G. contributed to meta-analysis of transcriptome data and helped writing the manuscript. C.S. contributed to experiments and provided insight. H.v.E. helped initiate the project and supported the research.

## Competing interests

The authors declare no competing interests.
