## [Peer Review File · Nature Communications]

Reviewers' Comments:

Reviewer #1:

Remarks to the Author:

In this manuscript, Xin Lei and colleagues investigated the transcriptomic signature of cDC1 licensed by CD4+ T cells in a human in vitro setting. They demonstrated that cDC1 can relay CD4+ T cells help for anti-tumor CTL priming, which is reflected in CD8+ T cells proliferation and Granzyme B induction. Then they showed the similarity of transcriptomic signature of "helped" cDC1 with that of clinically favorable DC states derived from different cancers. The "helped" cDC1 state was also correlated with CTL- and TH1 cell infiltration in TME by bioinformatics analysis. Overall, this study is largely descriptive, the mechanism of CTL priming by helped cDC1 is absent.

Q1: The cell number of pDC co-cultured with active CD4+ T cells (32 cells) or naïve CD4+ T cells (32 cells) is too small in scRNA-seq analysis. The transcriptomic analysis based on such a small cluster could be miss-leading, or at least not statistically significant to reflect the response of pDC to CD4+ T cells help.

Q2: The authors confirmed increased expression of many molecules at the protein level in cDC1 after co-culture with active CD4+ T cells (figure 2a and ,2b), but they used only cDC1 as control. cDC1 co-cultured with naïve CD4+ T cells is a needed control to demonstrate the function of activated CD4+ T-cells.

Q3: The authors found that CD70 on cDC1 is important for the CTL response to CD4+ T-cell help in human (figure 3), which was based on known finding in the mouse. To go a step further than published finding, if they directly activated CD70 receptor (CD27) on CD8+ T cells, can cDC1 partially induce CTL response without CD4+ T cells help?

Q4: The DC cells and CD4+ T cells used for scRNA-seq are all from PBMCs of healthy donor without any tumor-related antigen. Can the signature of helped cDC1 based on the in vitro system mimic cDC1 licensed by CD4+ T cells in a complex TME? And the signature of helped cDC1 is broader than that of DC3 or DC_S3. Is it possible that only a subtype of cDC1 is licensed by CD4+ T cells in TME?

Minor points:

1. The p value in figure 2d is absent.
2. The annotation of colors in extended figure 1h is absent.

Reviewer #2:

Remarks to the Author:

In this manuscript by Lei and colleagues, entitled "CD4+ T-cell help explains clinical benefit of specific intra- tumoral dendritic cell states in human cancer", the authors provided novel biological insights about the cellular and molecular pathways by which activated CD4 T cells provides "help" signals to the cDC1 dendritic cells, thus leading to greater CTL priming and effector functions. In a comparative analysis, the authors performed RNAsequencing of different DC subsets, ex vivo and after co-culture with activated or naïve CD4+ T cells, identifying a gene signature in "helped" cDC1 that was associated with antigen presentation and T cell co-stimulation. The authors validated these findings at the protein level and using challenging co-culture experiments. In addition, they provided mechanistic insights of the role of CD70, expressed by the helped-cDC1, as an important co-stimulatory molecule driving the greater CTL effector functions. Importantly, the authors identified a similar gene signature as the one observed in helped-cDC1 in several public NGS datasets from patients with cancers and after PD-1 blockade therapy, which associated with better outcome.

Overall, this work provides interesting and novel findings which are highly relevant to the field of cancer immunology. However, several important weaknesses hinder clear interpretation of the findings and do not allow strong conclusions to be drawn.

Indeed, while the systematic comparison between major DC subsets from human peripheral blood is highly relevant to the scientific question, this comparison cannot be readily made here because

of specific concerns that are listed below, and which should all be considered and addressed by the authors at this stage:

pDC datasets:

1. Can the authors explain why there are so few pDCs (32 cells) detected and/or analyzed in the RNAseq analysis, as compared to the other DC subsets (ranging from 150 to 445 cells), as indicated in ED Fig 3?
2. In addition, in Fig1a and in ED Fig.2d, a substantial number of cDC1 cells (HTO2 and 6 in red and orange respectively) are present within the pDC cluster (and some cDC2 cells in the cDC1 cluster). Considering the initial low number cells in the pDC cluster, this is a major concern that hinder further interpretation of the pDC data.
3. It appears that only two independent experiments with independent donors were used in the functional assays investigating the role of pDCs (Fig2, ED Fig6), which does not allow statistical analysis of the data. These experiments should be strengthened by additional replications.
4. The above points (low number of cells, clustering discrepancies, low number of repetition and absence of statistical analysis) may certainly affect the quality of the analysis and thus the conclusions that could be drawn. The authors should provide additional transcriptomic and functional datasets using pDCs in order to conclude on the role of these cells in their experimental settings.

moDC biology:

5. Could the authors provide a documented and literature-based rational of why they used CD206+ monocytes (called in their manuscript "MoDCs"), instead of GM-CSF-stimulated monocytes which are widely used in the literature as "moDCs"? How the CD206 monocytes used here are similar to monocyte-derived DCs after GM-CSF-stimulation?
6. Considering the above point, the authors should reconsider the denomination of the CD206+ monocytes as "MoDC" in their manuscript, which could be misleading.
7. In any case, the authors should backup their findings using moDCs from GM-CSF-stimulated monocytes as these DCs have been widely used to interrogate DC-Tcell communication, as well as in clinical applications.

Additional points:

8. As the CTL effector functions relate on cytotoxic activity, proliferation and cytokine production, it would be great to extend the readouts provided in Fig2 and ED Fig6 (CTV dilution and Granzyme B expression) by intracellular stainings of TNFa and IFNg.
9. The CD206 staining, as provided in Extended Data Fig.1a, does not seem optimal. What are the validation assays used for this antibody?
10. Extended Data Fig. 2: An additional tSNE plot, same as the one in ED Fig. 2d, but with a color mapping based on the three different donors should be also provided in order to evaluate inter-individual variability.
11. Line 47: "Among these, cDC1 and cDC1 are discerned into migratory..."

Reviewer #3:

Remarks to the Author:

This study builds on data and paradigms largely established in mouse models concerning the role of CD4+ T cell help in augmenting cDC1-mediated CD8+ T cell priming. The major advance delivered is the translation of this concept to humans and in particular the identification of a signature specifically associated with cDC1 after co-culture with polyclonal activated CD4+ T cells. This signature correlates with DC populations associated with prognosis and responses to immunotherapy in human tumor specimens leading to the major conclusion that CD4+ T cell help provided specifically to cDC1 is important for clinical benefit. Although the data provides robust evidence that the "help" signature induced by activated CD4+ T cells is induced in cDC1 and not other DC subsets it is not clear whether this constitutes a unique "help" as opposed to a more general "activation" DC signature that could also be induced by other innate / cytokine stimuli.

There are several concerns in the experimental design and interpretation of the findings as outlined below.

Major points:

1. Definition and gating strategy for MoDC is not aligned with current phenotyping markers used to define DC subsets in human peripheral blood (eg Villani Science 2017, Dutertre Immunity 2019). Hence the relevance of the data concerning this subset is questionable.
2. The cDC1 "help" signature appears very similar to what would be expected from activation by innate stimuli or cytokines. The "help" signature should therefore be compared to signatures associated with activation of cDC1 by innate stimuli to confirm whether this is a unique or more common signature of cDC1 activation.
3. According to the current paradigm (p3, lines 50-58), CD4+ T cells "license" cDC1 after antigen uptake and proinflammatory signals. In this study the activated CD4+ T cells are added to non-antigen bearing, immature DC. How this relates to the physiological context should be discussed.
4. How important is antigen / MHC II interactions or the lack thereof in this study likely to contribute to cDC1 mediated help?
5. Were the MART1 specific CD8+ T cells generated from the same donors as the DC and CD4+ T cells? It isn't stated whether these donors were also confirmed HLA-A2+.
6. Given the high precursor frequency of naïve MART1 specific CD8+ T cells in healthy donors direct priming of naïve MART1 CD8+ T cells could be examined against this antigen. This would be a more appropriate setting to investigate CD8+ T cell priming as opposed to activation of memory cells.
7. Several publications have demonstrated efficient cross-presentation of soluble or tumour antigen by human cDC1 in the absence of additional CD4+ T cell help. Can the authors please comment on why this does not appear to be the case in their model? Do the DC subsets pulsed with short peptide induce CD8+ T cell proliferation? Other measures of effector function(eg IFN γ) and phenotype should also be examined.
8. The Mel526 tumour cell line is A2+MART1+ and therefore has potential to also stimulate CD8+ T cells directly or via cross-dressing, which is also mediated by cDC1 (eg MacNabb et al Immunity 2022). A2-MART1+ and MART1- cell lines should be used to exclude these possibilities and support the statements that enhanced CTL priming is mediated by cross-presentation. Enhanced proliferation in the tetramer negative population (Fig 2e, lines 170-172) could also be due to non-specific and/or allogeneic responses. Granzyme B expression (Fig 2F) should be analysed in the tetramer pos and neg populations separately.
9. Was CD70 amongst the DEG genes or only upregulated as surface protein?

Lei et al. point-by-point reply

Reviewer #1 (Remarks to the Author):

In this manuscript, Xin Lei and colleagues investigated the transcriptomic signature of cDC1 licensed by CD4+ T cells in a human in vitro setting. They demonstrated that cDC1 can relay CD4+ T cells help for anti-tumor CTL priming, which is reflected in CD8+ T cells proliferation and Granzyme B induction. Then they showed the similarity of transcriptomic signature of “helped” cDC1 with that of clinically favorable DC states derived from different cancers. The “helped” cDC1 state was also correlated with CTL- and TH1 cell infiltration in TME by bioinformatics analysis. Overall, this study is largely descriptive, the mechanism of CTL priming by helped cDC1 is absent.

Q1: The cell number of pDC co-cultured with active CD4+ T cells (32 cells) or naïve CD4+ T cells (32 cells) is too small in scRNA-seq analysis. The transcriptomic analysis based on such a small cluster could be miss-leading, or at least not statistically significant to reflect the response of pDC to CD4+ T cells help.

Reply: We thank the reviewer for raising this concern. Indeed, the number of pDC that delivered data in the scRNA-Seq analysis is very low as compared to the other DC subsets, even though we entered the same amount of cells for each DC subset in the experiment. Loss is likely due to washing steps and/or the quality filtering steps (cell survival, multiplets). We have mentioned this in the Results section, line 116.

However, the inability of pDC to relay help for the CTL response, as concluded from our study, is not primarily based on the transcriptomics, but on the functional studies we have performed. In these studies, we sorted cells of all primary DC subsets and entered them in equal numbers in the experiments for side-by-side comparison.

In the revised manuscript, we have increased the robustness of the functional studies with pDC by adding additional samples from multiple independent donors for side-by-side comparison with the other DC subsets. The current data allow us to conclude with statistical significance that 1) upon incubation with activated CD4+ T cells, pDC did not alter protein expression of any of the key markers for DC activation/cross-presentation that are part of the “help” signature, in contrast to cDC1 (new Figure 2a and new Extended Data Figure 5g, Results section lines 134-137). 2) pDC did not relay CD4+ T-cell help for CTL priming (revised Figure 2c-e, see Figure legend for new n number and statistics).

Q2: The authors confirmed increased expression of many molecules at the protein level in cDC1 after co-culture with active CD4+ T cells (figure 2a and ,2b), but they used only cDC1 as control. cDC1 co-cultured with naïve CD4+ T cells is a needed control to demonstrate the function of activated CD4+ T-cells.

Reply: We agree with this relevant point and now present the results of a new experiment in which we compared side-by-side for n=3 the protein expression of key molecules that are part of the “help” signature in all DC subsets stimulated with naïve or activated CD4⁺ T cells (new Figure 2a, Extended Data Figure 5g, Results section lines 134-137).

Q3: The authors found that CD70 on cDC1 is important for the CTL response to CD4⁺ T-cell help in human (figure 3), which was based on known finding in the mouse. To go a step further than published finding, if they directly activated CD70 receptor (CD27) on CD8⁺ T cells, can cDC1 partially induce CTL response without CD4⁺ T cells help?

Reply: This data is the first demonstration of the importance of CD70 on cDC1 for relaying CD4⁺ T cell help for the CTL response in *human*. Since TNF family members and their receptors are highly regulated and can take over each other’s function, we believe that it contributes to the field to show the mouse to human conservation of this important feature of CD70. As we cite in the paper, it is known from mouse studies that apart from CD27 costimulation, also CD28 costimulation and cytokine signals are important downstream of “help” signals. Bypassing “help” by CD27 agonism is best revealed from our experience *in vivo* and with a deep read out such as transcriptomics (Ahrends *et al.* Immunity 2017). Nevertheless, we have performed an experiment as requested and present the results here for the reviewer. To stimulate human CD27, we used recombinant mouse FcCD70 that cross-reacts with human CD27. For n=2, we found a trend that FcCD70 promoted the ability of cDC1 to induce a CTL response in absence of CD4⁺ T-cell help. However, we prefer not to add this data in the main body of the manuscript since they are preliminary and not key for our main conclusions.

Effect of CD27 agonism in the CD8⁺ T cell priming system. On day 1, sorted HLA-A2⁺ cDC1 were incubated with activated (a) or naïve (n)CD4⁺ T-cells or without CD4⁺ T cells. MART-1₁₅₋₄₀ long peptide was added or not. At day 2, CTV-labeled CD8⁺ T-cells transduced with the MART-1₂₆₋₃₅/HLA-A2-specific TCR were added and validated soluble recombinant CD70 with Fc tail (FcCD70) in the setting where CD4⁺ T-cells were absent. (a) Flow cytometry plots depicting MART-1-specific CD8⁺ T-cell proliferation based on CTV dilution and intracellular Granzyme B staining. (b) Quantification of % MART-1₂₆₋₃₅/HLA-A2-specific (tetramer⁺) cells within CTV-negative (-) CD8⁺ T-cells (left) and the number (#) of live MART-1-specific CTV-CD8⁺ T-cells (right). (c) Quantification of % Granzyme B⁺ cells among CD8⁺ T-cells (left) and MFI of Granzyme B expressed by CD8⁺ T-cells (right). Data are pooled from two independent experiments (n=2), each with technical duplicates. Data are shown as means ± standard deviation (SD).

Q4a: The DC cells and CD4+ T cells used for scRNA-seq are all from PBMCs of healthy donor without any tumor-related antigen. Can the signature of helped cDC1 based on the in vitro system mimic cDC1 licensed by CD4+ T cells in a complex TME?

Reply: With our study, we identify the transcriptomic signature of cDC1 licensing by CD4+ T cells *in vitro*, using CD4+ T cells and DC subsets from human blood. In our setting, we use CD4+ T cells and DC regardless of their HLA type and in absence of antigen. By activating the CD4+ T cells via the TCR/CD3 complex with anti-CD3 antibody, we mimic recognition of the MHC/peptide complex. It is known that upon TCR/CD3 signaling, the T cell forms an immunological synapse with the antigen-presenting cell, based on integrin activation and resulting cell adhesion, which is supported by CD28 signaling that we also induce with specific antibody (Acuto & Michel *Nat. Rev. Immunol.* 2003). During the ensuing synaptic interaction, receptor-ligand communication between the activated T cell and the DC occurs. We now explain this scenario in the Results section lines 104-107.

We subsequently show the relatedness of the cDC1 “help” signature as obtained *in vitro* with published signatures of tumor-infiltrating DC, namely the DC3 state (Gerhardt *et al.* *J. Exp. Med.* 2021), the IL-32^{hi} intra-melanoma DC state (Gruber *et al.* *JCI Insight* 2020) and the positively prognostic DC_S3 state found in >6000 tumors of 16 different tissue types (Luca *et al.* *Cell* 2021). From these findings, we suggest that we can indeed find evidence for cDC1 licensing by CD4+ T cells in the TME. However, the CD4+ T cell help signature in cDC1 can potentially also be induced by other DC activating stimuli, the exact constellation of which in the TME we cannot predict. We have added data in Extended Data Figure 6 on this point in reply the reviewer 3 and present these in the Results on lines 134-142 and relate to them in the Discussion on lines 286-288.

Q4b: And the signature of helped cDC1 is broader than that of DC3 or DC_S3. Is it possible that only a subtype of cDC1 is licensed by CD4+ T cells in TME?

Reply: The reported gene signature of DC3 (47 genes) (Gerhardt *et al.* 2021) and DC_S3 (222 genes) (Luca *et al.* 2021) were obtained from the respective papers and related links. We cannot conclude that the signature of “helped” cDC1 is broader than that of DC3 and DC_S3, since full gene lists of these tumor-infiltrating DC are not available. cDC1 are considered a lineage; in that sense we are not aware of intrinsically different subtypes of cDC1 in the TME. However, in this environment these cells will likely attain different functional states that are reflected by their transcriptome depending on the exact micro-niche they find themselves in. We have now included these points in the revised Discussion section, lines 286-288.

Minor points:

1. The *p* value in figure 2d is absent.

Reply: The experiment in Figure 2d was for n=2, but based on the reviewer's important comment, we have carried out additional experiments with cells from multiple donors and now present statistical analysis for all DC subsets in revised Figures 2c-e.

2. *The annotation of colors in extended figure 1h is absent.*

Reply: Thank the reviewer for pointing this out; we have added the annotation of colors.

Reviewer #2 (Remarks to the Author):

In this manuscript by Lei and colleagues, entitled "CD4+ T-cell help explains clinical benefit of specific intra-tumoral dendritic cell states in human cancer", the authors provided novel biological insights about the cellular and molecular pathways by which activated CD4 T cells provides "help" signals to the cDC1 dendritic cells, thus leading to greater CTL priming and effector functions.

In a comparative analysis, the authors performed RNAsequencing of different DC subsets, ex vivo and after co-culture with activated or naïve CD4+ T cells, identifying a gene signature in "helped" cDC1 that was associated with antigen presentation and T cell co-stimulation. The authors validated these findings at the protein level and using challenging co-culture experiments. In addition, they provided mechanistic insights of the role of CD70, expressed by the helped-cDC1, as an important co-stimulatory molecule driving the greater CTL effector functions. Importantly, the authors identified a similar gene signature as the one observed in helped-cDC1 in several public NGS datasets from patients with cancers and after PD-1 blockade therapy, which associated with better outcome.

Overall, this work provides interesting and novel findings which are highly relevant to the field of cancer immunology. However, several important weaknesses hinder clear interpretation of the findings and do not allow strong conclusions to be drawn.

Indeed, while the systematic comparison between major DC subsets from human peripheral blood is highly relevant to the scientific question, this comparison cannot be readily made here because of specific concerns that are listed below, and which should all be considered and addressed by the authors at this stage:

Reply: We thank the reviewer for the appreciation of our study regarding the mechanistic insights obtained, their relevance for the field and the challenges taken on in our approach. We also thank the reviewer for the excellent feedback which has helped us to improve the manuscript.

pDC datasets:

Q1. Can the authors explain why there are so few pDCs (32 cells) detected and/or analyzed in the RNAseq analysis, as compared to the other DC subsets (ranging from 150 to 445 cells), as indicated in ED Fig 3?

Reply: As shown in Extended Figure 2a, all DC subsets were initially plated with equal cell numbers. We may have lost pDC during washing steps before 10x super-loading and we may have lost pDC information as a result of data filtering after scRNA-Seq analysis (multiplets, data quality). We now state this on line 113. Given the fact that the pDC count is low, we agree that we cannot make strong statements regarding the impact of CD4⁺ T-cells on pDC based on scRNA-Seq analysis. Instead, we base our conclusions on the functional follow-up experiments, for which we have now increased the number of samples and thereby the statistical robustness, particularly of the pDC data (see Q3).

Q2. In addition, in Fig1a and in ED Fig.2d, a substantial number of cDC1 cells (HTO2 and 6 in red and orange respectively) are present within the pDC cluster (and some cDC2 cells in the cDC1 cluster). Considering the initial low number cells in the pDC cluster, this is a major concern that hinder further interpretation of the pDC data.

Reply: Indeed, we also noticed that a small number of pDC are mixed and analyzed as cDC1 cells based on their hashtag identity. To address this issue, we have also annotated DC subsets according to their mRNA expression instead of their hashtag identities. This method increased the number of pDC in our analysis (about 30 more cells), but the number of pDC remained low compared to other DC subsets. Using mRNA expression for annotation did not identify any DEGs when comparing pDC co-cultured with naïve versus activated CD4⁺ T-cell conditions. So the overall outcome was similar when using mRNA expression or hashtag to annotate DC subsets. See further reply to Q1.

Q3. It appears that only two independent experiments with independent donors were used in the functional assays investigating the role of pDCs (Fig2, ED Fig6), which does not allow statistical analysis of the data. These experiments should be strengthened by additional replications.

Reply: We agree with this important comment. In the revised manuscript, we have increased the robustness of the functional studies with pDC by adding additional samples from multiple independent donors for side-by-side comparison with the other DC subsets. The current data allow us to conclude with statistical significance that 1) upon incubation with activated CD4⁺ T cells, pDC did not alter protein expression of any of the key markers for DC activation/cross-presentation that are part of the “help” signature, in contrast to cDC1 (new Figure 2a and new Extended Data Figure 5g, Results section lines 134-137). 2) pDC did not relay CD4⁺ T-cell help for CTL priming (revised Figure 2c-e, see Figure legend for new n number and statistics).

Q4. The above points (low number of cells, clustering discrepancies, low number of repetition and absence of statistical analysis) may certainly affect the quality of the analysis and thus the conclusions that could be drawn. The authors should provide additional transcriptomic and functional datasets using pDCs in order to conclude on the role of these cells in their experimental settings.

Reply: Please also see response to Q1 and 3. Indeed, the number of pDC that delivered data in the scRNA-Seq analysis is very low as compared to the other DC subsets. However, the inability of pDC to relay help for the CTL response, as concluded from our study, is not primarily based on the transcriptomics, but on the functional studies we have performed.

moDC biology:

Q 5. *Could the authors provide a documented and literature-based rationale of why they used CD206+ monocytes (called in their manuscript “MoDCs”), instead of GM-CSF-stimulated monocytes which are widely used in the literature as “moDCs”? How the CD206 monocytes used here are similar to monocyte-derived DCs after GM-CSF-stimulation?*

Q6. *Considering the above point, the authors should reconsider the denomination of the CD206+ monocytes as “MoDC” in their manuscript, which could be misleading.*

Q7. *In any case, the authors should backup their findings using moDCs from GM-CSF-stimulated monocytes as these DCs have been widely used to interrogate DC-Tcell communication, as well as in clinical applications.*

Reply: We thank the reviewer for alerting us that the term monocyte-derived DC (moDC) can refer to DC derived from monocytes *in vivo* or upon culture *in vitro*. We have followed the flow cytometric marker definitions of Durand & Segura, Front Immunol 2015 (PMID: 25852695) that diagnose “inflammatory” monocyte-derived DC directly *ex vivo* from human blood (CD14⁺CD11c⁺CD206⁺). We have now changed the Introduction on lines 45-49 to better explain the origin of moDC and progenitor-derived DC and adapted the references to reflect the development of the nomenclature. We have added new data to show that the moDC as we isolate them from blood have indeed the characteristics of moDC as identified in multiple publications. This is already apparent from the marker expression as shown in Extended Data Figure 1f and becomes clear from the new comparative transcriptomic analyses we now present in the new Extended Data Figure 2a-c, where we have compared our *ex vivo* moDC with myeloid cell types identified by leading single cell studies (Villani *et al.* Science 2017, PMID: 28428369; Dutertre *et al.* Immunity 2019, PMID: 31474513). In addition, we show that our *ex vivo* moDC largely share the gene expression signature of *in vitro* derived moDC (Balan *et al.* J. Immunol. 2014, PMID: 25009205). See Results section lines 89-100.

Furthermore, we now show that *in vitro* derived moDC do not respond to activated CD4⁺ T cells and in contrast to cDC1 cannot relay help for CTL priming in our *in vitro* assay of tumor antigen cross-presentation (new Extended Data Figure 10). See Results section lines 179-182. In this way, we have clarified that our results reflect the function of moDC in general.

Additional points:

Q8. *As the CTL effector functions relate on cytotoxic activity, proliferation and cytokine production, it*

would be great to extend the readouts provided in Fig2 and ED Fig6 (CTV dilution and Granzyme B expression) by intracellular stainings of TNF α and IFN γ .

Reply: We thank the reviewer for this suggestion. In new experiments for the revision, we have accordingly extended the read-out for CTL effector function. As now shown in the new Extended Data Figure 9, in the *in vitro* CTL priming assay with either long peptide or dead melanoma cell debris, “help” signals led to upregulation in responder CD8⁺ T cells of not only Granzyme B, but also TNF α , IFN γ , CD137 and CD45RO (n=2). See Results section lines 173-175.

Q9. The CD206 staining, as provided in Extended Data Fig.1a, does not seem optimal. What are the validation assays used for this antibody?

Reply: We thank the reviewer for this note. Indeed, CD206 expression is not detected at very high level *in ex vivo* moDC in Extended Data Fig.1a and it usually appears as a “smear” in conventional flow cytometry as used for cell sorting. In these sortings, we use cDC2 (within the same sample) as a negative control. However, in recent analyses, we have used spectral flow cytometry as shown in the new Extended Data Figure 10b. Here, one can see that the same antibody gives a clear positive staining on moDC as isolated *ex vivo* and generated *in vitro*.

Q10. Extended Data Fig. 2: An additional tSNE plot, same as the one in ED Fig. 2d, but with a color mapping based on the three different donors should be also provided in order to evaluate inter-individual variability.

Reply: It would indeed be interesting to know the inter-individual variability, but the experiment is not set up to deduce that information. We used 8 hashtags to distinguish four DC subsets co-cultured with either activated- or naïve CD4⁺ T-cells. Before hashtagging, we pooled each DC subset from three different donors to obtain enough cells for the single-cell experiment. Hence, we cannot track the information of the donors.

Q11. Line 47: “Among these, cDC1 and cDC1 are discerned into migratory...”

Reply: Text has been modified.

Reviewer #3 (Remarks to the Author):

This study builds on data and paradigms largely established in mouse models concerning the role of CD4+ T cell help in augmenting cDC1-mediated CD8+ T cell priming. The major advance delivered is the translation of this concept to humans and in particular the identification of a signature specifically associated with cDC1 after co-culture with polyclonal activated CD4+ T cells. This signature correlates with DC populations associated with prognosis and responses to immunotherapy in human tumor specimens leading to the major conclusion that CD4+ T cell help provided specifically to cDC1 is important for clinical benefit. Although the data provides robust evidence that the “help” signature induced by activated CD4+ T cells is induced in cDC1 and not other DC subsets. it is not clear whether this constitutes a unique “help” as opposed to a more general “activation” DC signature that could also be induced by other innate/cytokine stimuli. There are several concerns in the experimental design and interpretation of the findings as outlined below.

Reply: We thank the reviewer for the appreciation of our study and the excellent feedback which has helped us to improve our manuscript.

Major points:

Q1. Definition and gating strategy for MoDC is not aligned with current phenotyping markers used to define DC subsets in human peripheral blood (eg Villani Science 2017, Dutertre Immunity 2019). Hence the relevance of the data concerning this subset is questionable.

Reply: We thank the reviewer for alerting us that the term monocyte-derived DC (moDC) can refer to DC derived from monocytes *in vivo* or upon culture *in vitro*. We have followed the flow cytometric marker definitions of Durand & Segura, Front Immunol 2015 (PMID: 25852695) that diagnose “inflammatory” monocyte-derived DC directly *ex vivo* from human blood (CD14⁺CD11c⁺CD206⁺). We have now changed the Introduction on lines 45-49 to better explain the origin of moDC and progenitor-derived DC and adapted the references to reflect the development of the nomenclature. We have added new data to show that the moDC as we isolate them from blood have indeed the characteristics of moDC as identified in multiple publications. This is already apparent from the marker expression as shown in Extended Data Figure 1f and becomes clear from the new comparative transcriptomic analyses we now present in the new Extended Data Figure 2a-c, where we have compared our *ex vivo* moDC with myeloid cell types identified by leading single cell studies (Villani *et al.* Science 2017, PMID: 28428369; Dutertre *et al.* Immunity 2019, PMID: 31474513). In addition, we show that our *ex vivo* moDC largely share the gene expression signature of *in vitro* derived moDC (Balan *et al.* J. Immunol. 2014, PMID: 25009205). See Results section lines 89-100.

Furthermore, we now show that *in vitro* generated moDC do not respond to activated CD4⁺ T cells and in contrast to cDC1 cannot relay help for CTL priming in our *in vitro* assay of tumor antigen cross-

presentation (new Extended Data Figure 10). See Results section lines 179-182. In this way, we have clarified that our results reflect the function of moDC in general.

Q2. The cDC1 “help” signature appears very similar to what would be expected from activation by innate stimuli or cytokines. The “help” signature should therefore be compared to signatures associated with activation of cDC1 by innate stimuli to confirm whether this is a unique or more common signature of cDC1 activation.

Reply: We thank the reviewer for this important comment. We have now specified in the Discussion that from all studies in the mouse system, it is not yet clear whether CD4⁺ T cell licensing gives cDC a unique gene expression and functional profile as compared to input by innate stimuli and cytokines. Rather, thus far CD4⁺ T cell input is thought to increment expression of molecules that are induced by the innate stimuli.

In response to the reviewer’s comment, we have performed a new experiment in which we tested side-by-side for n=3 the protein expression of “help” signature proteins in cDC1 in response to either activated or naive CD4⁺ T-cells or the combination of the PRR stimuli (LPS, poly I:C and R848) that we also used in the cross-priming assay. In this setting, upregulation of costimulatory molecules appeared to be the common signature, whereas e.g. molecules involved in antigen (cross-)presentation appeared to be more responsive to “help” as compared to PRR stimulation (new Extended Data Figure 6 a-d). We describe these data in the Results section lines 134-142 and have added sentences to the Discussion on lines 283-285 and line 291.

Q3. According to the current paradigm (p3, lines 50-58), CD4+ T cells “license” cDC1 after antigen uptake and proinflammatory signals. In this study the activated CD4+ T cells are added to non-antigen bearing, immature DC. How this relates to the physiological context should be discussed.

Reply: We thank the reviewer for these questions. The first step of T cell priming is taken care of by migratory DC that very likely have seen PAMPs in tissue. However, there is evidence in the literature as reviewed in Borst *et al.* Nat. Rev. Immunol. 2018 that the second step of priming is taken care of by lymph node resident DC that may not have been in contact with PAMPs (or DAMPs). These DC are thought to acquire the antigen from the migratory DC after they have died. So our setting, in which we add activated CD4⁺ T cells to otherwise non-activated DC can be viewed as such a scenario. Likewise, contact between a CD4⁺ T cell and an immature DC can happen in the TME, where PAMPs are likely absent. We have chosen the setting to determine specifically the effect of activated CD4⁺ T cells on DC at the transcriptome and protein level. In the revised version of the manuscript, we compare it with the effect of PRR stimulation. In regards to the fact that our DC are non-antigen-bearing, please see response to Q4.

Q4. How important is antigen / MHC II interactions or the lack thereof in this study likely to contribute to cDC1 mediated help?

Reply: *In vivo*, antigen/MHC II recognition by the TCR of the CD4⁺ T cell will be the driving force for the communication between the CD4⁺ T cell and the DC. However, in our *in vitro* system we bypass the requirement for cognate interaction. By activating the CD4⁺ T cells via the TCR/CD3 complex with anti-CD3 antibody, we mimic recognition of the MHC/peptide complex. It is known that under these conditions, the T cell forms an immunological synapse with the antigen-presenting cell, based of TCR/CD3-mediated integrin activation and cell adhesion, which is supported by CD28 costimulation (Acuto & Michel, Nat. Rev. Immunol. 2003). During the ensuing synaptic interaction, receptor-ligand communication between the activated T cell and the DC occurs. We now explain this scenario in the Results section lines 104-107.

This is why the system works regardless of antigen/MHC II recognition by the TCR of the CD4⁺ T cell. With naïve CD4⁺ T cells, we can get some activation if the DC are not from the same donor. However, only a very small fraction of the CD4⁺ T cells in the pool used will recognize allo-MHC. Therefore, the comparative setting between activated and naïve CD4⁺ T cells is sufficiently discriminatory.

Q5. Were the MART1 specific CD8⁺ T cells generated from the same donors as the DC and CD4⁺ T cells? It isn't stated whether these donors were also confirmed HLA-A2+.

Reply: The MART-1 specific CD8⁺ T cells, as well as CD4⁺ T cells were not always from the same donors as the DC. We only confirmed that the donors of DC were HLA-A2 positive, since the DC have to present the MART-1 antigen in HLA-A2. We have now specified this point in the Methods section in the revised manuscript on lines 332-334.

Q6. Given the high precursor frequency of naïve MART1 specific CD8⁺ T cells in healthy donors direct priming of naïve MART1 CD8⁺ T cells could be examined against this antigen. This would be a more appropriate setting to investigate CD8⁺ T cell priming as opposed to activation of memory cells.

Reply: We thank the reviewer for this comment. We would like to emphasize that in the protocol we have developed, the TCR transduced CD8⁺ T cells attain a stem cell-like memory (T_{SCM}) state (Extended Data Fig. 7b, c). T_{SCM} cells are “semi-naïve” in the sense that they have not yet undergone any aspects of effector/memory or effector differentiation, according to transcriptome- and epigenetic analysis (Gattinoni *et al.* Nat. Med., 2017). Indeed, it would be more physiological to use naïve MART-1 specific CD8⁺ T cells for the priming assay. However, the precursor frequency of MART-1/HLA-A2-specific CD8⁺ T-cells is low and variable between donors. We chose to use MART-1/HLA-A2-specific TCR transduced T cells to have a stable and enlarged window for reading out CD8⁺ T-cell cross-priming. We have examined the response of naturally occurring MART-1/HLA-A2-specific CD8⁺ T-cells in our system that illustrates the small window for read out, as shown in the figure below.

Priming of naïve MART-1₂₆₋₃₅/HLA-A2-specific CD8⁺ T-cells by human cDC1. Purified cDC1 from HLA-A2⁺ healthy donors were incubated with activated- or naïve CD4⁺ T-cells and loaded with dead HLA-A2-MART-1⁺ Mel AAT cell debris. Next, cells were co-cultured with CTV-labeled naïve polyclonal CD8⁺ T-cells. **(a)** Gating strategies for detecting antigen specific CD8⁺ T-cell priming in our *in vitro* priming system based on CTV dilution (proliferation) and intracellular Granzyme B staining (CTL differentiation). **(b)** Flow cytometric plots depicting MART-1₂₆₋₃₅/HLA-A2-specific (tetramer⁺) CD8⁺ T-cell proliferation and Granzyme B staining. **(c)** Quantifications of % MART-1₂₆₋₃₅/HLA-A2-specific (tetramer⁺) cells, % CTV⁺ tetramer⁺ cells and % Granzyme B⁺ tetramer⁺ cells within CD8⁺ T-cells respectively (left panel) and quantifications of # MART-1₂₆₋₃₅/HLA-A2-specific (tetramer⁺) cells, # CTV tetramer⁺ cells and granzyme B expression (MFI) by tetramer⁺ cells (right panel). Data were pooled from two independent experiments (n=2) and shown as means ± SD.

Q7a. Several publications have demonstrated efficient cross-presentation of soluble or tumour antigen by human cDC1 in the absence of additional CD4⁺ T cell help. Can the authors please comment on why this does not appear to be the case in their model? Do the DC subsets pulsed with short peptide induce CD8⁺ T cell proliferation?

Reply: We thank the reviewer for this question. Our point is that CD4⁺ T cell help **increases** the capacity of cDC1 to induce CD8⁺ T cell cross-priming, on top of the contribution of PRR stimulation, as recapitulated in our priming assay. We have now incorporated data on the effects of PRR stimulation on the cDC1 that clarify that CD4⁺ T cell help enforces all elements required for cross-priming, but particularly antigen (cross)presentation (new Extended Data Figure 6, Results lines 134-142, Discussion lines 286-292).

In long-peptide cross-presentation setting, but not in the more challenging cell associated antigen cross-presentation, we see a MART-1 specific CD8⁺ T cell response in absence of help in case of the cDC1 (Figure 2c). We present here a figure that enlarges the window so that this response can be appreciated.

The cDC1 perform best in cross-priming as compared to the other DC types, also in absence of CD4⁺ T cell help (a,b). In this figure, we also show that all DC types could induce a response to MART-1 short peptide in the absence of CD4⁺ T cell help (c).

Priming by DC subsets in absence of help. Sorted HLA-A2⁺ pDC, cDC1, cDC2 and moDC were entered in the priming system with MART-1₁₁₅₋₄₀ long peptide (a, b) or MART-1₁₂₆₋₃₅ short peptide (c) without CD4⁺ T-cells. CTV-labeled CD8⁺ T-cells transduced with the MART-1₁₂₆₋₃₅/HLA-A2-specific TCR were used as responders. (a) Flow cytometry plots depicting MART-1-specific CD8⁺ T-cell proliferation based on CTV dilution and CTL differentiation based on intracellular Granzyme B staining (b) Quantification of the % responder CTV-negative (–) MART-1₁₂₆₋₃₅/HLA-A2-specific (tetramer⁺) cells within CD8⁺ T-cells and the % Granzyme B⁺ cells among CD8⁺ T-cells. Data were pooled from two independent experiments with (n=2). Data are shown as means ± SD. p<0.05*, p<0.01**, p<0.001*** (One way ANOVA). (c) Primary flow cytometry data of response to short peptide from one experiment with (n=1).

Q7b. Other measures of effector function (eg IFN γ) and phenotype should also be examined.

As now shown in new Extended Data Figure 9, in the *in vitro* CTL priming assay with either long peptide or dead melanoma cell debris, “help” signals led to upregulation in responder CD8⁺ T cells of not only Granzyme B, but also TNF α , IFN γ , CD137 and CD45RO (n=2). See Results section lines 173-175.

Q8a. The Mel526 tumour cell line is A2+MART1+ and therefore has potential to also stimulate CD8+ T cells directly or via cross-dressing, which is also mediated by cDC1 (eg MacNabb et al Immunity 2022). A2-MART1+ and MART1- cell lines should be used to exclude these possibilities and support the statements that enhanced CTL priming is mediated by cross-presentation.

Reply: When using HLA-A2+ cell line Mel 526 as MART-1 antigen source, we indeed cannot exclude that cross-dressing of the cDC1 with the peptide/MHC complex takes place. Therefore, we have now specified that we have also used MART-1+ HLA-A2- Mel AAT cells (new Extended Data Figure 7f) in these assays that give the same results. This is now presented in the new Extended Data Figure 9 and mentioned in the Results section lines 176-179. We have not used MART1- tumor cells, because then we do not have a read out for the CTV labeled HLA-A2/MART-1 TCR transduced CD8+ T-cells. To exclude direct recognition of the HLA-A2/MART-1 complex on dead melanoma cell debris, we present in new Extended Data Figure 9 g,h an experiment wherein DC were absent but dead Mel 526 or Mel AAT cell debris was present. No response of HLA-A2/MART-1 TCR transduced T-cells was observed.

Q8b. Enhanced proliferation in the tetramer negative population (Fig 2e, lines 170-172) could also be due to non-specific and/or allogeneic responses.

Reply: We agree that responses in the tetramer negative populations can also stem from responses to non-tumor related antigens. We have picked up responses to other HLA-A2-restricted melanoma antigens with specific tetramers, but prefer not to go into this in the present paper.

Q8c. Granzyme B expression (Fig 2F) should be analysed in the tetramer pos and neg populations separately.

Reply: Accordingly, we now depict analysis of Granzyme B expression in the tetramer positive- and negative populations separately (revised Extended Data Figure 8h-i).

Q9. Was CD70 amongst the DEG genes or only upregulated as surface protein?

Reply: We did not detect CD70 amongst the DEGs in our analysis.

Reviewers' Comments:

Reviewer #1:

Remarks to the Author:

The authors have experimentally addressed all of my major concerns, there are only a few minor issues left:

Q1: A small population of cDC1 cells in Fig.1a is shown in the pDC population (green circle) in extended data Fig.3d.

Q2: The cDC2 population in extended data Fig.5a is not a distinct cell population compared with the extended data Fig.1a.

Q3: The quality of figures in the revised manuscript could be improved, especially Fig.2a.

Reviewer #2:

Remarks to the Author:

In this revised version of the manuscript entitled "CD4+ T-cell help explains clinical benefit of specific intra- tumoral dendritic cell states in human cancer" the authors were responsive and addressed well the issues raised on the first submission. Most of the specific concerns were adequately addressed and the revised manuscript is of higher quality, which further strengthen the authors' conclusions and the novelty of the findings. Nevertheless, some issues remain and need to be addressed:

1- In this manuscript, moDC should be referred as "CD206+ moDC" or "ex vivo moDC", to avoid confusion with in vitro-generated moDC.

2- That the transcriptional signature profile of ex vivo moDC is similar to the DC3 subset (CD14+CD163+), as described in the study by Dutertre et al (ref 21), (ED Fig.2b) does not demonstrate "a unique monocyte-related signature" as stated Line 95, but is rather confusing. Indeed, one could expect more similarity between the DC3 signature in Dutertre et al and the Helped/licenced cDC1, as based on the authors findings: Line 76 -"We discovered the similarity of the "help" transcriptomic signature with recently identified DC3 and DC_S3 states in the TME". The authors should clarify this point, especially differences between blood and tissue DC3s.

3- ED Fig.1d ; the figure needs to be edited as it seems that a white square overlaps to the heatmap, at the top left.

Reviewer #3:

Remarks to the Author:

The authors have prepared a very considered and detailed response that adds significant new data and also adequately addresses the initial concerns. This is a high quality and significant body of work that will be of interest to the field.